# Towards Green AI in Fine-tuning Large Language Models via Adaptive Backpropagation

**Kai Huang[†], Hanyun Yin[§], Heng Huang[‡] & Wei Gao[†]**
University of Pittsburgh[†], University of Maryland, College Park[‡]
University of Science and Technology of China[§]
k.huang@pitt.edu, ykissgoodbye@gmail.com, heng@umd.edu, weigao@pitt.edu

## Abstract

Fine-tuning is essential to adapting pre-trained large language models to downstream applications. With the increasing popularity of LLM-enabled applications, fine-tuning has been performed intensively worldwide, incurring a tremendous amount of computing costs that correspond to big carbon footprint and environmental impact. Mitigating such environmental impact directly correlates to reducing the fine-tuning FLOPs. Existing fine-tuning schemes focus on either saving memory or reducing the overhead of computing weight updates, but cannot achieve sufficient FLOPs reduction due to their ignorance of the training cost in backpropagation. To address this limitation, in this paper we present *Green-Trainer*, a new technique that minimizes the FLOPs of LLM fine-tuning via adaptive backpropagation, which adaptively selects the most appropriate set of LLM tensors for fine-tuning based on their importance and backpropagation cost in training. Experiment results show that GreenTrainer can save up to 64% training FLOPs compared to full fine-tuning, without any noticeable accuracy loss. Compared to the existing schemes such as Prefix Tuning and LoRA, GreenTrainer can achieve up to 4% improvement of model accuracy, with on-par FLOPs reduction.

## 1 Introduction

Large language models (LLMs) are used as foundational tools in generative AI. To be used in downstream applications, a pre-trained LLM needs to be fine-tuned using the specific application data (Devlin et al., 2018). Intuitively, fine-tuning is less computationally expensive than pre-training due to the smaller amount of training data, but it may result in significantly high energy consumption and carbon footprint when being intensively performed worldwide. Enabled by the democratization of open-sourced LLMs (Candel et al., 2023) and convenient APIs of operating these LLMs (Ott et al., 2019; Wolf et al., 2019), even non-expert individuals can easily fine-tune LLMs for model performance enhancement or personalization (Scialom et al., 2022; Wang and Gao, 2023). For example, when a LLaMA-13B model (Touvron et al., 2023) is fine-tuned by 10k users using A100-80GB GPUs, such fine-tuning consumes $6.9\times$ more GPU hours than pre-training a GPT-3 model (Brown et al., 2020) with 175B parameters. The amount of energy consumed by such fine-tuning is comparable to that consumed by some underdeveloped countries, and the amount of carbon footprint is equivalent to $1000\times$ of that produced by a New York-San Francisco flight (aii, 2023).

Mitigating such environmental impact towards Green AI directly correlates to reducing the number of floating operations (FLOPs) of fine-tuning, which represents the amount of computational operations and hence energy consumption in training (Schwartz et al., 2020; Huang et al., 2023a). Most existing techniques of optimizing LLM fine-tuning, however, are limited to reducing the memory consumption rather than FLOPs (Malladi et al., 2023; Liao et al., 2023). Some other methods reduce FLOPs by only fine-tuning certain types of model parameters such as bias (Zaken et al., 2021), LayerNorm and output layer weights (Lu et al., 2021), but they impair the model's expressivity and are only applicable to simple non-generative learning tasks. Instead, researchers suggested keeping the original model parameters frozen but injecting additional trainable parameters to the input (Lester et al., 2021; Liu et al., 2022) or internal layers (Li and Liang, 2021; Hu et al., 2023; Huang et al., 2023b). Recent LoRA-based methods (Hu et al., 2021; Zhang et al., 2023) further reduce the overhead of computing weight updates for these injected parameters via low-rank approximation. These methods can minimize the model's accuracy loss on generative tasks. However, they still need to compute the activation gradients through the whole model and their FLOPs reduction is hence limited, because the computations of weight updates are only 25%-33% of the total training FLOPs.

Besides computing weight updates, FLOPs in training are also produced in i) forward propagation and ii) backward propagation of activation gradients. Since complete forward propagation is essential to calculate the training loss, we envision that the key to further FLOPs reduction is to take the backpropagation cost of activation gradients, which is >33% of the total training FLOPs, into account and selectively involve only the most appropriate model structures in backpropagation. The major challenge, however, is that selective training will possibly bring model accuracy loss. We minimize the accuracy loss is by adapting such selection in backpropagation to a flexible objective of FLOPs reduction, determined by the carbon footprint in energy supply. For example, when such carbon footprint is low due to insertion of renewable energy, using a lower objective of FLOPs reduction can involve more model structures in training and retain the training accuracy. High carbon footprint, instead, leads to a higher objective of FLOPs reduction for better embracing Green AI.

In this paper, we present *GreenTrainer*, a new technique that realizes adaptive backpropagation for efficient LLM fine-tuning with the minimum accuracy loss. As shown in Figure 1, given an objective of FLOPs reduction, GreenTrainer adaptively selects the set of trainable neural network (NN) tensors in each epoch, based on evaluation of tensors' importance in training. Such importance evaluation is difficult because NN tensors do not directly associate with input data variables or intermediate features, and most attribution techniques (Sundararajan et al., 2017; Hesse et al., 2021) that evaluate feature importance are not applicable. Popular importance metrics, including SNIP (Lee et al., 2018) and Fisher (Liu et al., 2021), are mainly used in NN pruning to quantify the importance of model weights at their current values, but they cannot quantify the importance of weight updates on a tensor to reducing the training loss. Classic metrics based on exact accuracy contribution (Lin et al., 2022), weight updates' magnitudes (Li et al., 2016), or random perturbations (Breiman, 2001), on the other hand, are either inaccurate or computationally expensive for LLMs. Instead, our approach adopts a similar rationale with the existing attribution and pruning metrics, and quantifies the contribution of each tensor update to the training loss via first-order Taylor expansion over the training loss. In this way, we ensure that the selected tensors can make the maximum contribution to reducing the training loss.

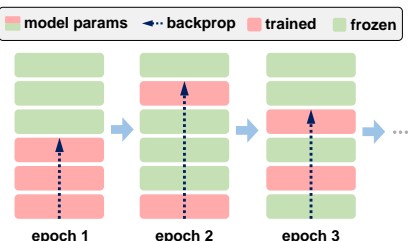

Figure 1: GreenTrainer adaptively selects the most appropriate portion of LLM model for fine-tuning

Another challenge is how to precisely profile the training FLOPs. Due to interdependency between tensors, their total FLOPs in training is not equal to the summation of their individual FLOPs. Such interdependency is determined by the backpropagation characteristics of NN operators in each tensor, but existing FLOPs models cannot link NN operators to tensors based on the computing flow of backpropagation. Some existing work (Kwon et al., 2023) only incorporates the layer-wise forward FLOPs into tensor selection, but ignores the computation dependency between layers in backpropagation. To tackle this challenge, we rigorously model the cross-tensor dependencies in profiling their backpropagation FLOPs. Based on this model, we develop a dynamic programming (DP) algorithm to find the nearly optimal tensor selection from an exponential number of possibilities (e.g., $2^{515}$ for 515 tensors in OPT-2.7B model (Zhang et al., 2022)). Therefore, GreenTrainer can make sure that the given objective of FLOPs reduction can be met in most cases.

We evaluated GreenTrainer with three open-sourced LLMs, namely OPT (Zhang et al., 2022), BLOOMZ (Muennighoff et al., 2022) and FLAN-T5 (Chung et al., 2022), on text generation datasets including SciTLDR (Cachola et al., 2020) and DialogSum (Chen et al., 2021). Our results show that GreenTrainer can save up to 64% training FLOPs compared to full LLM fine-tuning, without any noticeable accuracy loss. In some cases, GreenTrainer can even improve the model accuracy compared to full fine-tuning, by removing model redundancy and overfitting. Compared to existing techniques such as Prefix Tuning (Li and Liang, 2021) and LoRA (Hu et al., 2021), GreenTrainer improves the model accuracy by 4% with the same amount of FLOPs reduction, and also provides users with the flexibility to balance between the training accuracy and cost depending on the needs of Green AI.

## 2 BACKGROUND & MOTIVATION

### 2.1 TRANSFORMER ARCHITECTURES FOR TEXT GENERATION

Current LLMs are stacked by transformer blocks (Vaswani et al., 2017), each containing a Multi-Head Attention (MHA) layer, LayerNorms (Ba et al., 2016), and a Feed-Forward Network (FFN).

Given an input sequence $X \in \mathbb{R}^{n \times d}$ with $n$ tokens, the MHA projects tokens into a $(Q, K, V)$ space $h$ times, using $h$ suites of trainable projectors $(W_Q^{(i)}, W_K^{(i)}, W_V^{(i)})_{i=1,...,h}$. Each projection $f_i :$ $\mathbb{R}^{n \times d} \to \mathbb{R}^{n \times \frac{d}{h}}$ is defined as $Q_i, K_i, V_i = XW_Q^{(i)}, XW_K^{(i)}, XW_V^{(i)}$. The output $(Q_i, K_i, V_i)$ then performs attention mechanisms to produce $O_i$ by weighting $V_i$ with the attention scores between $Q_i$ and $K_i$. The MHA's final output is obtained by concatenating each $O_i$, following a linear projection $g : \mathbb{R}^{n \times d} \to \mathbb{R}^{n \times d}$ with a trainable projector $W_o$:

$$O_i = \text{Softmax}\left(Q_i K_i^\top / \sqrt{d/h}\right) V_i, \qquad \text{MHA}_{\text{out}} = \text{Concat}(O_1, O_2, ..., O_h)W_o. \quad (1)$$

To improve the training efficiency, LLMs adopt the teacher-forcing method (Lamb et al., 2016) to generate the entire sequence of output tokens in a single forward pass. Specifically, causal masks are applied to MHA's attention scores, so that each output token can be predicted from the label tokens at previous positions. With this technique, when being fine-tuned, LLMs can be trained in a standard way like any feed-forward models.

## 2.2 The Need for Adaptive Backpropagation

When being fine-tuned for a downstream task, LLMs are usually over-parameterized, because only part of the world knowledge that they learned from pre-training is useful for the target task. In these cases, only involving some of the model's substructures into fine-tuning could have little impact on the model accuracy, but significantly reduces the amount of computations.

| Trainable substructure | OPT-2.7B | | FLAN-T5-3B | |
|---|---|---|---|---|
| | **FLOPs** $(\times 10^{15})$ | **Acc. (%)** | **FLOPs** $(\times 10^{15})$ | **Acc. (%)** |
| All params | 262.0 | 23.6 | 135.7 | 46.5 |
| Last 2 layers | 181.6 (31%↓) | 20.8 | 46.1 (66%↓) | 39.2 |
| Decoder prefix | 174.7 (33%↓) | 13.4 | 55.3 (60%↓) | 37.6 |
| $(W_Q, W_V)$ | 174.7 (33%↓) | 23.8 | 90.5 (33%↓) | 44.7 |

Table 1: Fine-tuning different substructures of OPT-2.7B and FLAN-T5-3B LLMs on the Dialog-Sum dataset (ROUGE-1 score on the test set is used as the accuracy metric)

Existing work has made attempts with fixed selections of some NN components, such as the last 2 layers, decoder prefixes (Li and Liang, 2021), and linear projectors $(W_Q, W_V)$ (Hu et al., 2021), in fine-tuning. However, due to the interdependencies of NN parameters (Jin et al., 2020), such fixed selections will significantly impair the model accuracy. As shown in Table 1, solely fine-tuning either the last 2 layers or decoder prefixes leads to up to 10% accuracy drop. The reason is that nearby NN substructures with interdependencies on the fixed selections are excluded from fine-tuning, and hence become inconsistent with those selected substructures. Increasing the density of selection, such as including all the linear projectors $(W_Q, W_V)$, could mitigate the model accuracy loss, but can save at most 33% FLOPs due to backpropagating activation gradients through transformer blocks. Naive methods of dynamic selections, such as expanding the trainable portion from the last layer, have the similar limitation.

The deficiency of these existing methods motivates us to enforce more flexible and adaptive selection of LLM substructures in backpropagation. In GreenTrainer, we develop a tensor importance metric that incorporates parameter dependencies to evaluate how fine-tuning each tensor contributes to the trained model's accuracy at runtime. Knowledge about such tensor importance, then, allows us to achieve the desired FLOPs reduction while maximizing the model accuracy.

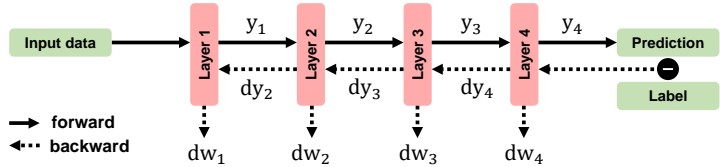

Figure 2: Backpropagation of a 4-layer dense NN

## 2.3 FLOPs Model of Backpropagation

The design of GreenTrainer relies on proper calculation of the selected model substructures' backpropagation FLOPs, which can be decomposed into two parts using the chain rule. For example, as

shown in Figure 2, when training a 4-layer dense NN without bias, each layer computes *i)* $dy_i$ as the loss $L$'s gradient w.r.t the activation $y_i$, and *ii)* $dw_i$ as the loss gradient w.r.t weight $W_i$, such that

$$dy_i = \frac{\partial L}{\partial y_i} = \frac{\partial L}{\partial y_{i+1}} W_i^\top = dy_{i+1} W_i^\top, \qquad dw_i = \frac{\partial L}{\partial W_i} = y_i^\top \frac{\partial L}{\partial y_{i+1}} = y_i^\top dy_{i+1}, \quad (2)$$

and the corresponding amounts of FLOPs for computing $dy_i$ and $dw_i$ are $t_{dy_i}$ and $t_{dw_i}$, respectively.

$(dy_i, dw_i)$ can be computed from $(dy_{i+1}, dw_{i+1})$. In particular, even if a layer is not selected in fine-tuning, it still needs to compute and pass error gradients $(dy_i)$ to the downstream layers. Hence, the amount of computations in backpropagation does not only depend on the selected layers, but also depends on some unselected layers. For example, if only Layer 2 is trainable, the total FLOPs for backpropagation will be decided by the cost of computing $dw_2$, $dy_3$ and $dy_4$. Due to the generality of the chain rule, such rationale of FLOPs calculation is also applicable to other types of NN layers.

Based on this rationale, we can construct FLOPs models for LLM substructures. The layer-level model is coarse-grained and can lead to inaccurate tensor selection. Some important parameters may be unselected due to other unimportant ones in the same layer. In GreenTrainer, we use tensor-level granularity for such selection, which can be well-supported by tensorized NN libraries (e.g., TensorFlow (Abadi, 2016) and PyTorch (Paszke et al., 2019)). Weight-level selection, although more fine-grained, is too computationally expensive due to the requirement of fine-grained indexing.

## 3 METHOD

To reduce the FLOPs of LLM fine-tuning, an intuitive problem formulation is to minimize the FLOPs while achieving the desired model accuracy. However, it is hard to determine a proper accuracy objective in advance, because some accuracy objectives may require very intensive training and the accuracy that we can achieve with our FLOPs budget cannot be pre-estimated before training. Instead, we maximize the training loss reduction while achieving the desired FLOPs reduction:

$$\max \Delta_{loss}(\boldsymbol{m}) \quad \text{s.t.} \ T_{selective}(\boldsymbol{m}) \leq \rho T_{full}, \quad (3)$$

where $\boldsymbol{m}$ is a binary vector to be solved for tensor selection. $\boldsymbol{m}$ parameterizes both the loss reduction ($\Delta_{loss}$) and per-batch FLOPs of training ($T_{selective}$), and $T_{selective}$ is constrained within a user-specified ratio ($\rho$) of the FLOPs of fine-tuning the whole model ($T_{full}$). For example, $\rho = 0.5$ means that the FLOPs of fine-tuning should be at most 50% of that in fine-tuning the whole model. In practice, the value of $\rho$ can either be preset or adjusted at runtime in any stage of training.

To identify each tensor's contribution in fine-tuning, we model $\Delta_{loss}(\boldsymbol{m})$ as the aggregated importance of selected tensors, and calculate the FLOPs incurred by selected tensors using the FLOPs model of backpropagation in Section 2.3. With this model, Eq. (3) can be rewritten as:

$$\max \Delta_{loss}(\boldsymbol{m}) \quad \text{s.t.} \ T_{fp} + \boldsymbol{m} \cdot \boldsymbol{t}_{dw} + \sigma(\boldsymbol{m}) \cdot \boldsymbol{t}_{dy} \leq \rho T_{full}, \quad (4)$$

where $T_{fp}$ indicates the per-batch FLOPs of the forward pass, and each pair of variables in $(\boldsymbol{t}_{dy}, \boldsymbol{t}_{dw})$ represents the FLOPs of computing $(dy, dw)$ for the corresponding tensor, respectively. Given a binary selector $\boldsymbol{m}$, $\sigma(\boldsymbol{m})$ incorporates all the tensors along the backward pass that contribute to the FLOPs of fine-tuning, by involving in passing the error gradients $(dy)$. For example, if $\boldsymbol{m} = [0, 0, 1, 0, 1, 0, 0]$, all the tensors that are in deeper layers than the selected tensors are involved in passing the error gradients, and hence $\sigma(\boldsymbol{m}) = [0, 0, 1, 1, 1, 1, 1]$.

To ground this formulation and solve $\boldsymbol{m}$, GreenTrainer consists of three key components: *(i) Tensor FLOPs Profiling*, which calculates the FLOPs of all NN tensors (i.e., $\boldsymbol{t}_{dy}$ and $\boldsymbol{t}_{dw}$) prior to training; *(ii) Tensor Importance Evaluation*, which quantifies the contribution of updating each NN tensor to the training quality at runtime; *(iii) Tensor Selector*, which grounds the tensor selection problem using tensors' FLOPs and importances, and provides solutions via dynamic programming at runtime.

### 3.1 TENSOR FLOPS PROFILING

Standard NN profilers, such as Torch Profiler (Paszke et al., 2019), can measure the execution FLOPs of individual NN operators such as matrix multiplication and convolution. However, it cannot be directly linked to NN tensors that participate in these operations. When a set of tensors is trained, the training FLOPs of backpropagation are not equal to the summation of individual tensors' FLOPs.

To address this limitation, our approach consists of two steps. First, we convert the layer-based NN structure of LLMs into a tensor-level computing graph, which retains the execution order of all

Figure 3: An sample workflow of tensor FLOPs profiling

tensors' involvements in training. Then, we extract the related backpropagation operators of each tensor, and derive each tensor $i$'s FLOPs in backpropagation ($t_{dy_i}$ and $t_{dw_i}$) by matching and aggregating the FLOPs of these NN operators. For example in Figure 3, the training of each linear projector ($Q$, $K$ and $V$) in an MHA layer should be executed after its corresponding bias tensor's training. Training each linear projector, then, will involve two matrix multiplication operators, whose FLOPs in backpropagation will be aggregated. We categorize such rules of matching and aggregation by the type of LLM layers where tensors are located, as described below. A specific example about such tensor FLOPs profiling on the OPT-2.7B model is provided in Appendix A.3.

**Input & output embedding layers.** The input embedding layer contains a trainable embedding tensor that maps each raw token into a dense representation. Given the activation gradient $dy_{i+1}$ from upstream layers, deriving the update of this tensor only involves variable assignment, and we can safely consider $t_{dw_i} \approx 0$ for any tensor $i$. If a raw token is mapped to the $k$-th vector in the embedding tensor during the forward pass, then during backpropagation, $dy_{i+1}$ from the upstream will be only assigned to $k$-th row of $dw_i$, such that $dw_i[s] = dy_{i+1}$ if $s = k$, otherwise $dw_i[s] = 0$. Since the input layer doesn't propagate activation gradients, we can also conclude that its $t_{dy}$ is 0.

Reversely, the output embedding layer projects each token back to the probability space. Intuitively, its $(t_{dy}, t_{dw})$ can be derived in the same way as we did for the dense layer in Eq. (2). However, in most LLMs, the output embedding layer shares the same trainable tensor with the input embedding layer. This implies that if the output embedding is trainable, then the input embedding will also be involved in training. Hence, all the $t_{dy}$ from LLM's output, up to the input embedding layer, should be accumulated to $t_{dy}$ of the output embedding tensor, while its $t_{dw}$ remains unchanged.

**Multi-Head Attention (MHA) layer.** An MHA layer contains multiple linear projectors as trainable tensors, and their FLOPs in training can be derived in the same way as we did with the dense layer in Eq. (2). Some LLMs (e.g., OPT) also include bias as another type of trainable tensor after such projection. In this case, based on the chain rule, the backpropagation of bias is computed as $dy_i = dy_{i+1}$ and $dw_i = 1^\top dy_{i+1}$, indicating that $t_{dy}$ for bias is 0 since $dy_i$ is identically passed from $dy_{i+1}$. $t_{dw}$ of bias can be derived as the FLOPs of adding up elements in $dy_{i+1}$ along every feature channel. The attention mechanism in Eq. (1) is backpropagated prior to the projectors. If any of these projectors are involved in training, the attention's backpropagation FLOPs must be also calculated, and we accumulate such FLOPs to the corresponding projector tensor ($W_V$)'s $t_{dy}$.

**LayerNorm.** Given a token, LayerNorm first normalizes its features and uses two trainable tensors $\gamma$ and $\beta$ to element-wise multiply with and add to the token, respectively. The operations of multiplication and addition are similar to those in the dense layer, and so its FLOPs can be calculated in the similar way. However, the backpropagation FLOPs of normalization operators should be accumulated to the previous tensor's $t_{dy}$. If any tensors in the previous layers are trained, the FLOPs of propagating the normalization operators should be also included in the FLOPs of the current layer.

**Feed-Forward Network (FFN).** In the FFN, there is a nonlinear activation function between two dense layers. Following the same method of calculating LayerNorm's FLOPs, we accumulate the FLOPs of propagating through this activation function to the bias tensor's $t_{dy}$ in the first dense layer.

### 3.2 TENSOR IMPORTANCE EVALUATION

A tensor's importance in training can be estimated as the summation of the importances of all its weights. In training, since the model weights are iteratively updated to minimize the training loss, an intuitive approach to evaluating the importance of a weight update in a given iteration is to undo this update and check how the training loss increases back as $\Delta L = L(w) - L(w + \Delta w)$, so that a higher value of $\Delta L$ means this update is more important and the weight should be selected. However, computing $\Delta L$ for every weight is expensive. Instead, we estimate the importance of all weights in one shot by smoothing the undo operation described above and computing the loss gradients with respect to the updates that correspond to all the weights. Letting the multiplicative

$c \in [0, 1]^M$ denote the undo operation for all the $M$ weights, we can compute the loss gradient as

$$-\frac{\partial L(w + c \odot \Delta w)}{\partial c} = -\Delta w \odot \left.\frac{\partial L(u)}{\partial u}\right|_{u=w+c\odot\Delta w}, \quad (5)$$

where $\odot$ denotes element-wise multiplication. When $c = 0$, Eq. (5) becomes an importance vector over all weights. Since the loss gradient is parameterized by all weights, the weight importances calculated in this way implicitly incorporate the impact of weight dependencies. A tensor $k$'s importance is then calculated as

$$I_k = -\sum_i \Delta w_i^{(k)} \partial L / \partial w_i^{(k)}. \quad (6)$$

In some cases, when the training process encounters divergence, the values of gradients and calculated tensor importances in Eq. (6) could be very large, eventually leading to overflow when using these importance values for deciding tensor selection in Eq. (4). To address this issue, we could further scale all the tensor importance by the maximum amplitude to improve numerical stability.

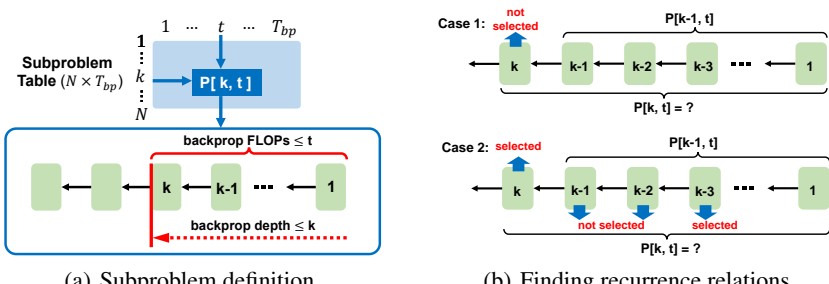

(a) Subproblem definition      (b) Finding recurrence relations

Figure 4: Solving the tensor selection problem using DP

## 3.3 TENSOR SELECTION

Since Eq. (4) is a nonlinear integer programming problem and hence NP-hard, in GreenTrainer we seek for an approximate solution using dynamic programming (DP). We decompose the whole problem into subproblems constrained by different depths of backpropagation. These subproblems can be sequentially solved from the one with the smallest depth, by using their recurrence relations.

**Subproblem definition.** As shown in Figure 4(a), we define each subproblem $P[k, t]$ as to maximize the cumulative importance of selected tensors when 1) selection is among the top $k$ tensors[1] and 2) backpropagation FLOPs is at most $t$. DP starts by solving the smallest subproblem $P[k = 1, t = 1]$ and gradually solves larger subproblems based on the results of smaller subproblems and the recurrence relation of these subproblems, until the target problem $P[N, T_{full}]$ is solved.

**Recurrence relations of subproblems.** The recurrence relation between subproblem $P[k, t]$ and $P[k - 1, t]$ depends on whether we further select the top tensor $k$ from the solution of $P[k - 1, t]$, as shown in Figure 4(b). **Case 1:** If $k$ is not selected, $P[k, t]$ will fall back to $P[k - 1, t]$, since the importance of selected tensors will not be further increased. **Case 2:** If $k$ is selected, then its FLOPs will be included into the solution of $P[k, t]$, no matter which other tensors are selected. The FLOPs involved with tensor $k$ include 1) the FLOPs to update tensor $k$ and 2) the FLOPs to pass activation gradients from the closest selected tensor $k_c$, such as tensor $k - 3$ as shown in Figure 4(b), to tensor $k$. This implies that $P[k, t]$ falls back to a previously solved subproblem $P[k - k_c, t - \Delta t]$, where

$$\Delta t = t_{dw_k} + \sum_{j=k_c}^{k-1} t_{dy_j}. \quad (7)$$

Since $k_c$ is unknown in advance, we backtrace the previously solved subproblems and explore all the possibilities of $k_c$ by reducing the depth of backpropagation from $k$, and the optimal solution to $P[k, t]$ is the one with the highest cumulative importance of selected tensors. Based on this recurrence relation, we can solve all subproblems by traversing the subproblem space. The time complexity of solving each subproblem is $O(N)$, and the overall time complexity of DP is $O(N^2 T_{full})$.

## 4 EXPERIMENTS

In our evaluation, we include decoder-only LLMs including OPT (Zhang et al., 2022) and BLOOMZ (Muennighoff et al., 2022), and an encoder-decoder LLM, namely FLAN-T5 (Chung et al., 2022),

---

[1]We consider the tensor that is closest to the NN output as the topmost.

with LLM sizes ranging from 350M to 6.7B. Our experiments are mainly conducted using the following two datasets of abstractive summarization:

- **SciTLDR** (Cachola et al., 2020) is a dataset of 5.4K text summaries on 3.2K papers. It contains both author-written and expert-derived TLDRs, where the latter is collected by an annotation protocol that produces high-quality summaries with low annotation burden.
- **DialogSum** (Chen et al., 2021) is a dialogue summarization dataset of 13,460 dialogues with manually labeled summaries and topics. It has been demonstrated more challenging than other summarization datasets, such as SAMSum (Gliwa et al., 2019) and CNN/Daily (Nallapati et al., 2016) at a similar scale.

We also perform generative QA tasks on WebQuestion (Berant et al., 2013) and PIQA (Bisk et al., 2020) datasets in Appendix A.4. However, we do not consider non-generative tasks such as sentimental classification, entailment classification and extractive QA, because these tasks are too easy for LLMs and testing them with LLMs will result in exaggerated performance gain over the baseline.

For OPT and BLOOMZ, we follow GPT2-like prompt structures (Radford et al., 2019), "[source seq.] TL;DR:", for summarization tasks to preprocess input data. For FLAN-T5, we adopt the prompt structure "summarize: [source seq.]" used in the original T5 pre-training. We truncate the source sequences so that the length of every preprocessed input sequence is within 512 tokens. On the test data, we use a beam search size of 4, and set the maximum number of generated tokens to 64 for SciTLDR and 128 for DialogSum. We compare GreenTrainer (GT) with the following baselines:

- **Full Fine-Tuning (Full FT)** fine-tunes all the LLM parameters and should intuitively achieve the best accuracy of the trained model.
- **Fine-Tuning Top2 (FT-Top2)** only fine-tunes the last two layers, typically the embedding layer and a LayerNorm. The input and output embedding layers are tied for OPT and BLOOMZ, but are not tied for FLAN-T5. This naive baseline only fine-tunes the smallest portion of LLM parameters and is used to identify whether the dataset is trivial to the LLM.
- **Prefix Tuning (Prefix-T)** (Li and Liang, 2021) inserts trainable prefixes into each transformer block's input sequence while freezing the model parameters. For encoder-decoder LLMs, the trainable prefixes are only inserted into the decoder blocks.
- **LoRA** (Hu et al., 2021) is currently the most popular method for efficient LLM fine-tuning. It uses low-rank matrix decomposition to reduce the training cost. We apply LoRA to both query and value projectors, as suggested in (Hu et al., 2021).

In all experiments, we use a batch size of 4 and fine-tune the model for 5 epochs. We use the AdamW optimizer (Loshchilov and Hutter, 2017) at a learning rate of $2 \times 10^{-5}$ with linear schedule and weight decay of $10^{-2}$. We use the ROUGE scores (%R1/R2/RL) (Lin, 2004) as the accuracy metric, and measure both Peta-FLOPs (PFLOPs) and wall-clock time as the training cost in each run. We measure the end-to-end cost of training, including the computing costs in forward and backward passes, and the computing costs of tensor importance evaluation and tensor selection using DP.

## 4.1 TRAINING COST & ACCURACY

We first evaluate the training cost and accuracy of GreenTrainer (GT). As shown in Table 2, for the OPT-2.7B model, GT-0.5 can achieve the required 50% of FLOPs reduction with at most 2% accuracy loss, and GT-0.7 can even achieve 0.2%-3% higher ROUGE scores than Full FT. We hypothesize that this is because GT only fine-tunes the most important tensors and hence mitigates the possible overfitting in Full FT. Insufficient trainable parameters can also lead to underfitting, as FT-Top2 has significantly lower ROUGE scores. Similarly, compared to LoRA and Prefix Tuning, GT-0.7 achieves at least 2% higher accuracy with the same amount of training FLOPs.

Similarly, for BLOOMZ-3B, GT-0.5 can save 50% training FLOPs and wall-clock time with $< 2\%$ accuracy loss. Compared to Full FT, GT-0.7 achieves the same ROUGE scores on SciTLDR, and 4%-10% higher on DialogSum. With the same amount of training FLOPs, GT-0.7 has 0.4%-1.4% higher ROUGE scores than LoRA. Note that both datasets are non-trivial for the BLOOMZ model, since the naive baseline (FT-Top2) still exhibits high accuracy loss.

For the FLAN-T5-3B model, FT-Top2 achieves similar fine-tuning qualities to Full FT with lower FLOPs, indicating that the SciTLDR dataset is trivial for FLAN-T5. In this case, GT-0.34 can achieve the same FLOPs and ROUGE scores by selecting a small portion of tensors. On the other hand, FT-Top2 loses accuracy significantly on DialogSum, but GT-0.4 reduces 54% of training FLOPs and 43% of wall-clock time without noticeable accuracy loss. GT-0.4 also outperforms LoRA by 1% on ROUGE scores and reduces 11% more FLOPs. Compared to Prefix tuning, GT-0.34 achieves 2%-5% higher ROUGE scores, while reducing the same amount of training FLOPs.

| # Model & Method | SciTLDR | | | DialogSum | | |
|---|---|---|---|---|---|---|
| | PFLOPs | Time (h) | R1/R2/RL | PFLOPs | Time (h) | R1/R2/RL |
| **OPT-2.7B** | | | | | | |
| Full FT | 41.8 | 0.92 | 32.9/14.9/27.1 | 262.0 | 5.5 | 23.6/9.5/18.8 |
| FT-Top2 | 29.0 (31%↓) | 0.61 (34%↓) | 9.1/4.0/7.6 | 181.6 (31%↓) | 3.8 (31%↓) | 20.8/7.9/17.5 |
| Prefix-T | 27.9 (33%↓) | 0.58 (37%↓) | 7.6/0.4/6.1 | 174.7 (33%↓) | 3.7 (33%↓) | 13.4/3.3/10.9 |
| LoRA | 27.9 (33%↓) | 0.59 (36%↓) | 28.2/12.1/21.0 | 174.7 (33%↓) | 3.6 (35%↓) | 23.8/9.5/18.8 |
| GT-0.5 | 20.8 (50%↓) | 0.46 (50%↓) | 30.5/13.1/25.2 | 130.1 (50%↓) | 2.7 (51%↓) | 21.4/8.2/17.6 |
| GT-0.7 | 29.2 (30%↓) | 0.68 (26%↓) | 33.1/15.2/27.6 | 182.7 (30%↓) | 4.0 (27%↓) | 26.8/11.0/21.6 |
| **BLOOMZ-3B** | | | | | | |
| Full FT | 47.2 | 1.0 | 28.3/12.1/22.5 | 294.8 | 6.5 | 26.1/10.6/21.0 |
| FT-Top2 | 36.5 (23%↓) | 0.75 (25%↓) | 23.7/8.8/18.8 | 227.9 (23%↓) | 4.6 (29%↓) | 22.1/8.5/17.8 |
| Prefix-T | 31.5 (33%↓) | 0.68 (34%↓) | 6.5/2.2/5.5 | 196.5 (33%↓) | 4.2 (35%↓) | 29.6/9.4/24.9 |
| LoRA | 31.5 (33%↓) | 0.69 (33%↓) | 27.4/11.7/21.8 | 196.5 (33%↓) | 4.3 (34%↓) | 35.4/14.3/28.6 |
| GT-0.5 | 23.4 (51%↓) | 0.51 (50%↓) | 26.7/10.7/21.2 | 146.4 (50%↓) | 3.1 (52%↓) | 24.9/9.5/20.0 |
| GT-0.7 | 32.3 (32%↓) | 0.74 (28%↓) | 28.0/12.2/22.4 | 204.7 (31%↓) | 4.3 (34%↓) | 36.8/14.7/29.4 |
| **FLAN-T5-3B** | | | | | | |
| Full FT | 21.7 | 0.64 | 37.1/18.5/31.7 | 135.7 | 4.0 | 46.5/20.8/38.5 |
| FT-Top2 | 7.3 (66%↓) | 0.21 (67%↓) | 36.5/18.4/31.5 | 46.1 (66%↓) | 1.4 (65%↓) | 39.2/16.7/32.9 |
| Prefix-T | 8.0 (63%↓) | 0.23 (64%↓) | 36.0/18.2/31.0 | 55.3 (60%↓) | 1.7 (57%↓) | 37.6/16.4/32.1 |
| LoRA | 14.4 (33%↓) | 0.41 (36%↓) | 36.6/18.5/31.5 | 90.5 (33%↓) | 2.5 (38%↓) | 44.7/19.8/37.1 |
| GT-0.34 | 7.5 (65%↓) | 0.23 (64%↓) | 36.4/18.4/31.7 | 53.5 (61%↓) | 1.4 (65%↓) | 42.7/18.3/35.1 |
| GT-0.4 | 10.0 (54%↓) | 0.38 (41%↓) | 36.7/18.5/31.5 | 62.5 (54%↓) | 2.3 (43%↓) | 46.0/20.7/38.1 |
| GT-0.5 | 12.4 (43%↓) | 0.44 (31%↓) | 36.3/17.7/30.9 | 77.6 (43%↓) | 2.6 (35%↓) | 46.2/20.7/38.1 |

Table 2: Comparison of the training cost & accuracy in LLM fine-tuning. GreenTrainer with an objective $\rho$ of FLOPs reduction is denoted as GT-$\rho$.

## 4.2 The Impact of FLOPs Reduction Objective

To better understand how GreenTrainer performs with different objectives of FLOPs reduction, we vary the value of $\rho$ between 0.36 and 0.8, and compare GreenTrainer with LoRA on the OPT-2.7B model. As shown in Table 3, on the SciTLDR dataset, when the requirement of FLOPs reduction is high and corresponds to a value of $\rho \leq 0.4$, GreenTrainer outperforms LoRA by achieving 2% higher ROUGE scores and saving 25% more FLOPs and wall-clock time. On the other hand, when the value of $\rho$ increases to 0.6, GreenTrainer outperforms the Full FT on ROUGE scores by 0.5% and outperforms LoRA by 5.2%, but saves 40% of training FLOPs and 39% of wall-clock time compared to Full FT. Similar results are also observed on the DialogSum dataset. In summary, with different objectives of FLOPs reduction, GreenTrainer can always provide better tradeoffs between the training accuracy and cost, compared to the SOTA baselines.

| Method | SciTLDR | | | DialogSum | | |
|---|---|---|---|---|---|---|
| | PFLOPs | Time (h) | R1/R2/RL | PFLOPs | Time (h) | R1/R2/RL |
| Full FT | 41.8 | 0.92 | 32.9/14.9/27.1 | 262.0 | 5.5 | 23.6/9.5/18.8 |
| LoRA | 27.9 (33%↓) | 0.59 (36%↓) | 28.2/12.1/21.0 | 174.7 (33%↓) | 3.6 (35%↓) | 23.8/9.5/18.8 |
| GT-0.36 | 14.9 (64%↓) | 0.32 (65%↓) | 4.1/1.7/3.6 | 92.9 (65%↓) | 1.9 (65%↓) | 15.7/5.0/13.8 |
| GT-0.4 | 16.6 (60%↓) | 0.36 (61%↓) | 28.6/11.6/23.5 | 103.4 (61%↓) | 2.2 (60%↓) | 17.9/6.3/15.4 |
| GT-0.5 | 20.8 (50%↓) | 0.46 (50%↓) | 30.5/13.1/25.2 | 130.1 (50%↓) | 2.7 (51%↓) | 21.4/8.2/17.6 |
| GT-0.6 | 25.0 (40%↓) | 0.56 (39%↓) | 33.4/15.3/27.8 | 156.6 (40%↓) | 3.3 (40%↓) | 24.0/9.7/19.2 |
| GT-0.7 | 29.2 (30%↓) | 0.68 (26%↓) | 33.1/15.2/27.6 | 182.7 (30%↓) | 4.0 (27%↓) | 26.8/11.0/21.6 |
| GT-0.8 | 33.4 (20%↓) | 0.77 (16%↓) | 33.1/15.5/27.6 | 209.6 (20%↓) | 4.4 (20%↓) | 23.9/9.9/19.1 |

Table 3: Impact of different objectives of FLOPs reduction on the OPT-2.7B model

These results also demonstrate that GreenTrainer provides great flexibility in LLM fine-tuning between the training accuracy and cost, by adjusting the value of $\rho$. The user can opt to set a low value of $\rho$ ($\leq 0.4$) to maximize the FLOPs reduction ($>60\%$) with moderate model accuracy loss (3%-4% on the two datasets we use). Alternatively, they can use a high value of $\rho$ ($\geq 0.6$) to have the same level of FLOPs reduction as that of LoRA, but ensure the minimum model accuracy loss or even minor model accuracy improvement. We believe that such flexibility is practically important when fine-tuning LLMs for downstream tasks with different green AI requirements and constraints.

## 4.3 Efficacy of Tensor Importance Metrics

The fine-tuning quality of GreenTrainer builds on proper evaluation of tensor importance. We compare our metric ($\Delta w \frac{\partial L}{\partial w}$) to the magnitude-based metric ($\Delta w$) (Lee et al., 2020) and the gradients-only metric ($\frac{\partial L}{\partial w}$) (Aji and Heafield, 2017), using the OPT-2.7B model with $\rho = 0.7$. As shown

| Method | SciTLDR | | | DialogSum | | |
|---|---|---|---|---|---|---|
| | PFLOPs | Time (h) | R1/R2/RL | PFLOPs | Time (h) | R1/R2/RL |
| Full FT | 41.8 | 0.92 | 32.9/14.9/27.1 | 262.0 | 5.5 | 23.6/9.5/18.8 |
| GT-0.7 ($\Delta w$) | 29.4 (30%↓) | 0.68 (26%↓) | 32.7/15.2/27.2 | 183.8 (30%↓) | 4.0 (27%↓) | 24.9/10.2/19.7 |
| GT-0.7 ($\frac{\partial L}{\partial w}$) | 29.4 (30%↓) | 0.67 (27%↓) | 32.8/15.1/27.2 | 184.0 (30%↓) | 4.0 (27%↓) | 25.0/10.2/20.0 |
| GT-0.7 ($\Delta w \frac{\partial L}{\partial w}$) | 29.2 (30%↓) | 0.68 (26%↓) | 33.1/15.2/27.6 | 182.7 (30%↓) | 4.0 (27%↓) | 26.8/11.0/21.6 |

Table 4: Efficacy of Tensor Importance Metrics (OPT-2.7B)

in Table 4, with the same objective of FLOPs reduction, using our metric ($\Delta w \frac{\partial L}{\partial w}$) for tensor importance evaluation achieves the highest model accuracy and outperforms Full FT by 1%-3% on ROUGE scores. This is because magnitude-based metrics ignore the dependencies of weight updates. Gradient-only metrics only contain the direction information about tensor importance but cannot reflect the intensity of importance. Inaccurate importance measurements will in turn lead to inappropriate selections of trainable tensors.

### 4.4 IMPACT OF LLM SIZE

A type of LLM may contain several variants with different sizes. To study GreenTrainer's performance with different LLM sizes, we performed fine-tuning using the OPT models with sizes ranging from 350M to 6.7B. As shown in Table 5, even on small models (OPT-350M), GT-0.5 can save 17%-21% more training FLOPs than LoRA does, while achieving 2%-4% higher accuracy (on SciTDR) or the same accuracy (on DialogSum). When the model size increases to 2.7B, GT-0.5 outperforms LoRA and GT-0.7 outperforms Full FT on the SciTLDR dataset. On DialogSum, GT-0.7 performs similarly compared to LoRA. For the OPT-6.7B model[2], GT-0.4 can save 27% more training FLOPs than LoRA does on SciTLDR, while achieving the same model accuracy, and similar advantages can also be observed when comparing GT-0.5 and GT-0.7 with LoRA. Generally speaking, Green-Trainer's performance advantage widely applies to LLMs with different sizes.

| # Params & Method | SciTLDR | | | DialogSum | | |
|---|---|---|---|---|---|---|
| | PFLOPs | Time (h) | R1/R2/RL | PFLOPs | Time (h) | R1/R2/RL |
| **OPT-350M** | | | | | | |
| Full FT | 5.4 | 0.15 | 30.9/13.9/25.7 | 33.8 | 0.92 | 23.2/9.0/18.5 |
| LoRA | 3.6 (33%↓) | 0.10 (33%↓) | 25.9/10.8/20.3 | 22.5 (33%↓) | 0.65 (29%↓) | 21.5/7.7/17.3 |
| GT-0.4 | 2.1 (61%↓) | 0.06 (60%↓) | 27.7/12.2/23.4 | 13.3 (61%↓) | 0.36 (61%↓) | 17.3/5.8/14.6 |
| GT-0.5 | 2.7 (50%↓) | 0.08 (47%↓) | 29.9/13.2/24.9 | 16.7 (51%↓) | 0.45 (51%↓) | 21.3/7.8/17.3 |
| GT-0.7 | 3.8 (30%↓) | 0.12 (20%↓) | 30.6/13.5/25.0 | 23.6 (30%↓) | 0.66 (28%↓) | 24.2/9.3/19.3 |
| **OPT-1.3B** | | | | | | |
| Full FT | 20.8 | 0.46 | 32.1/14.3/26.4 | 130.8 | 2.9 | 25.4/10.3/20.2 |
| LoRA | 13.9 (33%↓) | 0.31 (33%↓) | 28.1/11.9/22.0 | 87.2 (33%↓) | 1.9 (34%↓) | 24.6/9.9/19.4 |
| GT-0.4 | 8.2 (61%↓) | 0.18 (61%↓) | 28.9/11.9/23.8 | 51.4 (61%↓) | 1.1 (62%↓) | 16.9/5.7/14.6 |
| GT-0.5 | 10.3 (50%↓) | 0.23 (50%↓) | 30.0/12.7/24.5 | 64.2 (51%↓) | 1.4 (51%↓) | 20.1/7.4/16.7 |
| GT-0.7 | 14.5 (30%↓) | 0.34 (26%↓) | 31.2/14.2/25.8 | 90.8 (30%↓) | 2.0 (31%↓) | 24.4/9.7/19.4 |
| **OPT-2.7B** | | | | | | |
| Full FT | 41.8 | 0.92 | 32.9/14.9/27.1 | 262.0 | 5.5 | 23.6/9.5/18.8 |
| LoRA | 27.9 (33%↓) | 0.59 (36%↓) | 28.2/12.1/21.0 | 174.7 (33%↓) | 3.6 (35%↓) | 23.8/9.5/18.8 |
| GT-0.4 | 16.6 (60%↓) | 0.36 (61%↓) | 28.6/11.6/23.5 | 103.4 (61%↓) | 2.2 (60%↓) | 17.9/6.3/15.4 |
| GT-0.5 | 20.8 (50%↓) | 0.46 (50%↓) | 30.5/13.1/25.2 | 130.1 (50%↓) | 2.7 (51%↓) | 21.4/8.2/17.6 |
| GT-0.7 | 29.2 (30%↓) | 0.68 (26%↓) | 33.1/15.2/27.6 | 182.7 (30%↓) | 4.0 (27%↓) | 26.8/11.0/21.6 |
| **OPT-6.7B** | | | | | | |
| Full FT | 103.9 | 5.44 | 32.9/14.9/27.5 | 649.9 | - | - |
| LoRA | 69.3 (33%↓) | 1.3 | 28.4/12.3/22.7 | 433.3 (33%↓) | 8.1 | 24.9/10.2/19.4 |
| GT-0.4 | 41.2 (60%↓) | 0.9 | 28.9/11.8/23.4 | 257.9 (60%↓) | 5.2 | 19.7/7.0/16.3 |
| GT-0.5 | 50.8 (51%↓) | 1.1 | 30.1/13.0/24.8 | 331.4 (49%↓) | 6.7 | 21.8/8.5/17.3 |
| GT-0.7 | 74.8 (28%↓) | 1.4 | 33.1/15.3/27.7 | - | - | - |

Table 5: Impact of LLM's model size

## 5 CONCLUSION

In this paper, we present GreenTrainer, a new technique for LLM fine-tuning that allows efficient selection of trainable parameters via adaptive backpropagation, to ensure high training quality while minimizing the computation cost. GreenTrainer saves up to 64% training FLOPs compared to full fine-tuning without noticeable accuracy loss. Compared to the existing technique such as Prefix Tuning and LoRA, GreenTrainer improves the accuracy by up to 4% with the same FLOPs reduction.

---

[2]For the OPT-6.7B, Full FT and GT-0.7 with DialogSum have the out-of-memory issue on GPUs we use.

## ACKNOWLEDGMENTS

We would like to thank the anonymous reviewers and area chair for their comments and feedback. This work was supported in part by National Science Foundation (NSF) under grant number IIS-2205360, CCF-2217003 and CCF-2215042.

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

# A  APPENDIX

## A.1  REDUCING THE MEMORY USAGE OF TENSOR IMPORTANCE EVALUATION

Our approach to evaluating the importance of NN tensors in Section 3.2 requires caching all the previous model weights and the current gradients, in order to compute Eq. (6). However, doing so significantly increases the GPU memory consumption, especially for modern LLMs with billions of model weights. To reduce such GPU memory usage, we observe that our problem formulation in Eq. (4) will prevent tensors in early layers to be selected for training, due to the high costs of propagating their activation gradients in backpropagation. Hence, we could safely exclude these tensors from the trainable portion of LLM fine-tuning and save a significant amount of GPU memory. More specifically, the backpropagation during tensor importance evaluation can be early stopped at a certain tensor $k$, such that

$$\sum_{i=k-1,\dots,N} t_{dy_i} < \rho T_{full} \le \sum_{i=k,\dots,N} t_{dy_i}, \tag{8}$$

i.e., the cumulative FLOPs of all the tensors from 1 to $k$ just exceeds our objective of FLOPs reduction. As shown in Table 6, by applying such early stopping method, we could proportionally save GPU memory with respect to the value of $\rho$, as a smaller value of $\rho$ leads to smaller $k$ and the backpropagation can hence be stopped earlier. For example, when $\rho =$50%, 25% of GPU memory can be saved, and such saving could further increase to 50% when $\rho =$34%.

| Model | Full evaluation | Early-stop $\rho = 34\%$ | Early-stop $\rho = 40\%$ | Early-stop $\rho = 50\%$ | Early-stop $\rho = 60\%$ |
|---|---|---|---|---|---|
| OPT-2.7B | 10.8 | 5.5 | 6.5 | 8.1 | 9.7 |
| FLAN-T5-3B | 12.0 | 6.1 | 7.2 | 9.0 | 10.8 |

Table 6: GPU memory consumption (in GigaBytes) of tensor importance evaluation

## A.2  REDUCING THE COMPUTATIONAL COST OF DYNAMIC PROGRAMMING FOR TENSOR SELECTION

In our proposed dynamic programming (DP) approach for tensor selection in Section 3.3, due to the high volume of FLOPs in LLM fine-tuning, the value of $T_{full}$ could be very large. To reduce the computational cost of DP, we can reduce the subproblem space by skipping two types of subproblems: 1) **invalid ones**, whose FLOPs constraint $t$ exceeds the desired constraint ($\rho T_{full}$); 2) **redundant ones**, whose FLOPs to pass activation gradients to the maximally allowed depth ($k$) exceeds $t$. Our preliminary experiment show that, doing so on an OPT model with $\rho_{bp} = 50\%$ can reduce the number of subproblems by 5.5× without affecting the optimality of training.

| Model | $T_q = 1e1$ | $T_q = 1e2$ | $T_q = 1e3$ | $T_q = 1e4$ | $T_q = 1e5$ |
|---|---|---|---|---|---|
| OPT-2.7B | 0.02/64.1/32.0 | 0.04/47.6/30.1 | 0.64/49.8/30.7 | 7.5/50.0/30.9 | 76.5/50.0/30.9 |
| BLOOMZ-3B | 0.0001/33.3/9.30 | 0.007/45.7/25.2 | 0.21/49.5/27.2 | 2.3/49.8/27.1 | 25.3/50.0/27.1 |
| FLAN-T5-3B | 0.04/64.9/36.5 | 0.25/57.1/36.5 | 3.5/55.3/36.7 | 41.8/51.8/36.7 | 449/50.0/36.7 |

Table 7: The impact of DP resolution $T_q$ on fine-tuning OPT-2.7B, BLOOMZ-3B, and FLAN-T5-3B LLMs, on the SciTLDR dataset with $\rho = 50\%$. Each triplet [a/b/c] presents a) the percentage of wall-clock time incurred by DP compared to full fine-tuning, b) the percentage of FLOPs after reduction compared to full fine-tuning, and c) the testing ROUGE-1 score, respectively.

Besides, to further reduce the number of subproblems, we scale tensors' FLOPs ($t_{dw}, t_{dy}$) by multiplying a factor of $Z$:

$$\widetilde{t_{dw}} = \lfloor t_{dw} \cdot Z \rfloor, \quad \widetilde{t_{dy}} = \lfloor t_{dy} \cdot Z \rfloor, \tag{9}$$

where $Z = \frac{T_q}{T_{full}}$ reduces the backropagation FLOPs to a resolution of $T_q < T_{full}$. The overall time complexity of DP is then reduced to $O(N^2 T_q)$. On the other hand, such reduced resolution could increase the ambiguity in DP and affect the training quality. To investigate such tradeoff between the training quality and cost, we conducted preliminary experiments on multiple LLMs. Results

in Table 7 show that, for both OPT-2.7B and BLOOMZ-3B models, setting $T_q = 1e3$ reduces the DP overhead to $< 1\%$ without affecting the training quality. Similarly, for FLAN-T5-3B, choosing $T_q = 1e2$ can retain good training quality with negligible overhead. On the other hand, when $T_q$ is too small, the solution of DP could be inaccurate and hence result in ineffective reduction of the training FLOPs.

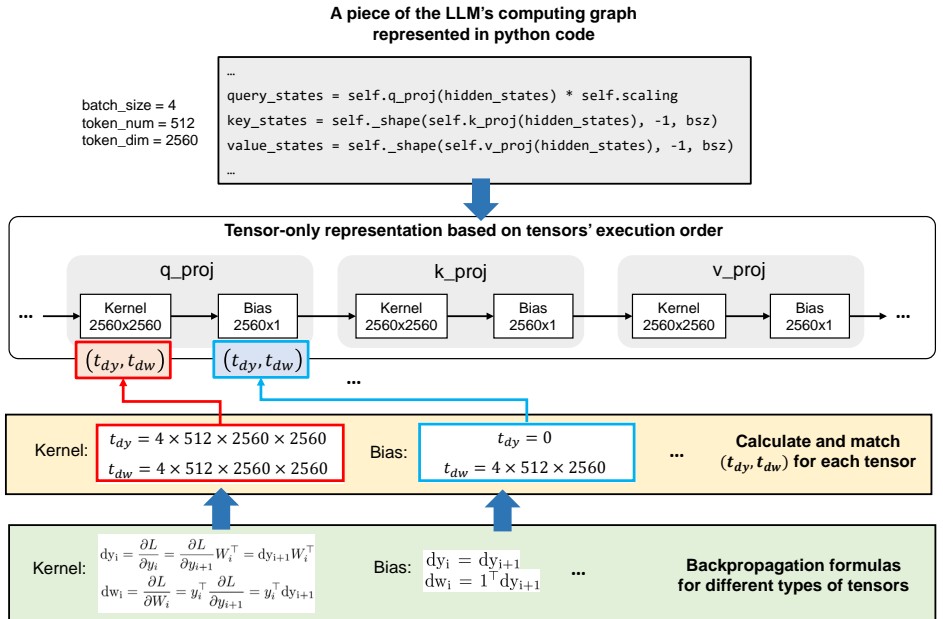

Figure 5: An example of tensor FLOPs profiling in the OPT-2.7B model

## A.3 AN EXAMPLE OF TENSOR FLOPS PROFILING IN THE OPT-2.7B MODEL

To better facilitate understanding, we further show an example in Figure 5 about how we profile tensors in the OPT-2.7B models in our experiments. First, we convert the computing graph of the LLM, which is implemented in Python code, into a tensor-only representation. The tensors are ordered based on their execution orders in the forward pass, similar to the layer-level graph in Figure 3. We then calculate each tensor's FLOPs $(t_{dy}, t_{dy})$ based on the backpropagation formulas discussed in Section 3.1. Such calculations are essentially counting the multiplications and being added in their formulas.

| Method | Accuracy (%) | PFLOPs | Time (h) |
|--------|-------------|--------|----------|
| LoRA | 49.5 | 174.0 | 6.27 |
| GT-0.5 | 59.2 | 130.5 | 4.69 |

Table 8: OPT-2.7B on PIQA dataset

| Method | Accuracy (%) | PFLOPs | Time (h) |
|--------|-------------|--------|----------|
| LoRA | 19.6 | 16.0 | 0.55 |
| GT-0.5 | 28.7 | 12.0 | 0.50 |
| GT-0.6 | 29.5 | 14.0 | 0.61 |

Table 9: OPT-2.7B on WebQuestion dataset

## A.4 PERFORMANCE ON GENERATIVE QUESTION-ANSWERING TASKS

To better evaluate the performance of GreenTrainer on other tasks, we also conducted experiments by using the OPT-2.7B model on WebQuestions and PIQA datasets for generative QA tasks. The WebQuestions dataset contains 6,642 QA pairs using Freebase as the knowledge base. The PIQA dataset focuses on multi-choice QA about physical knowledge with 21k QA pairs. We adopt the prompt format "`question:{q}answer:{a}`" for WebQuestions and "`goal:{q}sol1:{sol1}sol2:{sol2}label:{a}`" for PIQA, where `` is the EOS token for OPT models. The hyper-parameters for training are the same as the ones described in Section 4. We evaluate the sentence-level accuracy which requires the generated answer to exactly match the ground truth. Note that for PIQA, the generated tokens are still predicted from the entire dictionary of OPT embeddings instead of from the two choices: the first

or the second one. As shown in Table 8 and Table 9, on both datasets, GreenTrainer (GT) achieves significantly better accuracy and time efficiency compared to LoRA.

In particular, the results on the PIQA dataset are generally lower than those reported in Brown et al. (2020). The reason for this accuracy gap is that the way we use the OPT model to generate answers is more challenging than the setup in Brown et al. (2020). According to Section 2.4 in Brown et al. (2020), it formulates the PIQA task as a multi-choice QA task where the answer is drawn from a small and predefined candidate set (e.g., ["0", "1"]), by comparing the probability scores only over the candidate tokens. In comparison, we strictly cast the problem to open-ended generation, where the candidate set is unknown. In that case, generating correct answers can be more difficult, because the model could generate totally irrelevant answers and increase its chance of making mistakes.

## A.5 IMPACT OF FREQUENCY OF TENSOR IMPORTANCE EVALUATION

Our design of GreenTrainer, by default, evaluates the importance of tensors and select the set of trainable tensors based on such importance at the beginning of each training epoch. Using the technical approach described in Section 3.1, such tensor importance evaluation is very lightweight, and our experiment results show that the overhead of importance evaluation is only 0.2% on SciTLDR dataset and 0.01% on DialogSum dataset, with respect to the entire fine-tuning cost.

On the other hand, in certain cases, the tensor importances, calculated from the model gradient changes, could exhibit non-negligible differences within one epoch. In these cases, the flexible design of GreenTrainer will allow us to adaptively increase the frequency of tensor importance evaluation and the corresponding DP-based tensor selection. To demonstrate the impact of such more frequent tensor importance evaluation and DP-based tensor selection, we conducted extra experiments using OPT-2.7B model on the WebQuestions dataset and generative QA task, as shown in Table 10.

| Frequency of tensor importance evaluation | Accuracy (%) | Time (h) |
|---|---|---|
| Every 945 iterations (once per epoch) | 28.4 | 0.50 |
| Every 600 iterations | 28.5 | 0.54 |
| Every 400 iterations | 28.2 | 0.56 |
| Every 200 iterations | 27.5 | 0.64 |

Table 10: Impact of tensor importance evaluation frequency

The results show that: (1) More frequent tensor importance evaluation brings only very small improvement on task accuracy. Considering the randomness in different training trials, we believe that such accuracy improvement is negligible, and the accuracy could even drop down by 1% when the frequency of evaluation is very high (every 200 iterations). We believe that this is due to accumulation of tensor importance evaluation and tensor selection errors, which stem from the first-order approximation in the tensor importance metric and the approximate solution in DP. Another possible reason is that the tensor importances are calculated over the training dataset, and too frequent tensor importance evaluation may make the training process overfit to the training dataset. (2) The training cost steadily increases with the frequency of tensor importance evaluation. When the interval of evaluation reduces from 945 iterations to 200 iterations, the training time increases by 28%.

In summary, performing more frequent tensor importance evaluation within each epoch brings little improvement on the task accuracy but noticeably increase the training cost. We believe that the tensor importances being evaluated once in each epoch would be sufficiently accurate for appropriate selection of trainable tensors.

## A.6 THE NECESSITY OF DYNAMIC TENSOR SELECTION

If the LLM fine-tuning uses a fixed training dataset, it is possible that using a fixed tensor selection decided at the initial phase of training may not result in a significant model accuracy drop, compared to runtime tensor selection. However, in practical LLM fine-tuning scenarios, this assumption usually does not hold due to the following two reasons. First, in a lot of LLM fine-tuning scenarios, such as online learning and model personalization, the model is continuously retrained using online data, which is continuously generated at runtime with variant data distributions. Such variant data distributions will surely result in different importances of tensors through the training procedure and

hence require runtime tensor selection. Such online LLM fine-tuning scenarios recently become more and more popular, especially with the possibility of deploying LLMs onto user's personal mobile devices such as smartphones. Second, even for a fixed training dataset, it is also possible that the importances of some tensors may change as the training progresses. In these cases, dynamic tensor selection could improve the trained model accuracy. To verify this, we conducted additional experiments using the OPT-2.7B model on the WebQuestions dataset and generative QA task. As shown in Table 11, dynamic tensor selection could make non-negligible contributions to improving the task accuracy, with negligible increase of training cost.

| Strategy | Accuracy (%) | Time (h) |
|---|---|---|
| Fixed tensor selection only in the first epoch of training | 27.4 | 0.49 |
| Dynamic tensor selection, once in each epoch | 28.4 | 0.50 |
| More frequent tensor selection (5 times in each epoch) | 27.5 | 0.64 |

Table 11: Different strategies of tensor selection

Note that, such improvement of model accuracy would be dependent on the specific dataset and model being used, but these experiment results above demonstrated the necessity of runtime tensor selection. In addition, our experiment results also showed that such tensor importance evaluation and selection indeed incur very little extra computing overhead.

