# OpenReview forum: "Towards Green AI in Fine-tuning Large Language Models via Adaptive Backpropagation"
_ICLR.cc/2024/Conference — ICLR 2024 poster_

### Official Review · Reviewer_U1j2 · 2023-10-25

**Soundness:** 3 good
**Presentation:** 3 good
**Contribution:** 3 good
**Rating:** 6
**Confidence:** 4

**Summary:**

In this paper, the authors propose GreenTrainer, a method to reduce the FLOPs of LLM fine-tuning by selective back-propagation. GreenTrainer consists of three techniques: tensor FLOPS profiling, tensor importance evaluation, and tensor selection (for update). Experiments show that the proposed method can achieve similar accuracy while reducing up to 64% of training FLOPs, which makes training faster and reduces energy usage.

**Strengths:**

1. The paper is generally well-written and easy to follow.
2. The proposed method can reduce the training FLOPs more compared to existing efficient fine-tuning methods like LoRA.
3. The ablation studies are comprehensive, showing the effectiveness of the tensor selection mechanism and the scalability with larger models.

**Weaknesses:**

1. The template is highly altered to fit in more content. For example, the author section is removed, which may violate the policy.
2. The paper misses some references to existing work that uses similar techniques. For example, [a] also uses sparse update for the model and a similar optimization goal as Equation 5 to get the tensor selection scheme, which also considers the backward cost for weights and activations.  Other related works include [b, c]. The authors should discuss and differentiate from existing works.
3. There are some other efficient training methods focusing more on FLOPs reduction instead of parameter reduction like [d]. The authors may also want to include them as baselines since LoRA and Prefix-tuning are not designed for computation efficiency.
4. Despite the effective FLOPs reduction, I would expect the method to use more GPU memory compared to methods like LoRA due to the larger optimizer states (since more weights are updated). The authors may also discuss the memory usage compared to the baseline methods, and see whether it can be combined with LoRA to achieve reduced memory and FLOPs at the same time (e.g., only add LoRA to updated tensors).
5. The tensor importance estimation is based on the gradient information. Is it consistent throughout the training, or will the importance change?


[a] Lin et al., On-Device Training Under 256KB Memory
[b] He et al., Sensitivity-Aware Visual Parameter-Efficient Fine-Tuning
[c] Kwon et al., TinyTrain: Deep Neural Network Training at the Extreme Edge
[d] Evci et al., Head2Toe: Utilizing Intermediate Representations for Better Transfer Learning

**Questions:**

Please see the weakness section. Thanks!

---

> ### Author Response · Authors · 2023-11-16
> **Response to reviewer's comments and questions**
>
> Dear reviewer,
>
> Thank you so much for your efforts on reviewing our paper. Please find our responses to your comments and questions below.
>
> > The template is highly altered to fit in more content. For example, the author section is removed, which may violate the policy.
>
> Thank you for the comment. We have revised the paper to add the author section and make sure that it comply to the submission instructions.
>
> > The paper misses some references to existing work that uses similar techniques ...
>
> The main goal of MCUNetV3 [a] is to reduce the memory cost of training, so that the training can be executed on MCUs with extremely low memory. This goal is orthogonal to that of GreenTrainer, which is to speed up LLM fine-tuning. The methods of importance evaluation in [a] are also different from our method in GreenTrainer. From its problem formulation (see Eq. (5) in [a]), it adopts accuracy contributions as the importance metric and hence requires enumeration on the training dataset. This is acceptable for [a]’s problem scenario where both importance evaluation and trainable component selection are done offline. In contrast, in GreenTrainer we aim to enable runtime selection of trainable tensors, and hence we designed methods of tensor importance evaluation and tensor selection with very low computing costs for efficient runtime operations.
>
> Similarly, SPT [b] also does offline selection. It builds on top of the existing Parameter-Efficient Fine-Tuning (PEFT) methods (e.g., LoRA) and always selects the top-k most important parameters in each adapter for fine-tuning. Since existing PEFT methods typically insert adapters to every LLM block, SPT still requires propagating activation gradients throughout the LLM, which cannot guarantee that the given objective of FLOPs reduction can be met.
>
> TinyTrain [c] handcrafts a layer importance metric to combine accuracy, parameter counts, and computation cost into a single formula. It then offline selects the top-k important layers for fine-tuning online. Although it considers both the computing cost and accuracy, simply selecting top-k important layers cannot guarantee reduction of computing cost, because its importance metric doesn’t consider the computation dependency between layers in backpropagation. In comparison, GreenTrainer rigorously models the backpropagation cost for training each tensor and use DP for tensor selection. Our proposed importance metric also ensures additivity among importances of individual tensors, to ensure proper tensor selection.
>
> > There are some other efficient training methods focusing more on FLOPs reduction instead of parameter reduction like [d] ...
>
> Head2Toe [d] is designed for visual classification models, by inserting untrained classification heads for classification models such as ViT and ResNet50. However, when being applied to LLMs, due to the dependency between input and output embeddings of LLMs, it would be difficult to similarly insert an untrained output embedding layer for generative LLMs.
>
> > Despite the effective FLOPs reduction, I would expect the method to use more GPU memory compared to methods like LoRA due to the larger optimizer states (since more weights are updated) ...
>
> GreenTrainer could consume more memory than LoRA, because GreenTrainer is not optimized for reducing the memory usage. However, as the reviewer suggested, GreenTrainer can build upon LoRA and be used together with LoRA. In that case, the whole set of trainable components is the LoRA adapters being inserted in the LLM blocks. GreenTrainer then can be directly applied to select which LoRA adapter to fine-tune.
>
> The extra memory usage in GreenTrainer is mainly due to the tensor importance evaluation that requires caching all the model weights and current gradients. In Appendix A.1, we proposed techniques to further reduce such memory usage, by applying early stop strategies and excluding tensors in early layers from being involved in tensor selection. Results in Table 6 showed that we can further reduce the memory usage by 50%, approximately to be half of that in full-training.
>
> > The tensor importance estimation is based on the gradient information. Is it consistent throughout the training, or will the importance change?
>
> The importance of tensors will change over training. We hypothesize the change of tensor importance stems from the irregular trajectory of how the model parameters move on the loss surface during training. As a result, we perform tensor importance evaluation per epoch to capture such dynamics.
>
> In addition, our proposed tensor importance evaluation, by design, incurs only negligible computing overhead, because it only takes one extra training iteration to obtain gradients for all model parameters. To further verify such low overhead, we conducted extra experiments and the results show that, even when we repeatedly perform tensor importance evaluation in each training epoch, its overhead is only <0.2% of the entire fine-tuning cost.

---

> > ### Comment · Reviewer_U1j2 · 2023-11-22
> >
> > Thanks to the authors for the rebuttal, which addresses many of my concerns, and sorry for the late response.
> >
> > - Thanks for adapting the format to be more consistent with the original template.
> >
> > - Thanks for the discussion on the difference against existing work. One core difference seems to lie in the dynamic tensor selection throughout the training. However, there lacks an ablation study (if I do not miss it) to show how important the "dynamic" part is in this method. According to the reply to Reviewer P1Bv, higher importance estimation frequency does not improve accuracy (in the extreme case, it actually degrades the accuracy). Then how about just using the fixed subset of the tensor determined by the initial iterations for the training, which is more regular and better for efficiency? Will there be a noticeable accuracy gap?

---

> ### Author Response · Authors · 2023-11-20
> **Follow up on our previous response**
>
> Dear Reviewer,
>
> We are writing to kindly follow up with our previous response to your review comments, and would like to kindly enquire whether our response fully answered your questions and addressed your concerns to our paper. We will be more than happy to provide further details and responses for any additional questions you may have. Again, thank you so much for your review efforts and your insightful feedback about our paper!

---

> ### Author Response · Authors · 2023-11-22
> **Further clarifications**
>
> Dear Reviewer,
>
> Thank you so much for your response, and we are glad to see that most of your questions and concerns have been addressed. Regarding your question about using a fixed set of selected tensors determined at the initial stage of training, please find our further clarifications as below.
>
> We would like to clarify that the correlation between the trained model's accuracy and frequency of tensor importance evaluation (and tensor selection) mainly depends on how much the tensors' importances will vary. In our previous responses to other reviewers, we justified that such variance in one epoch would be very small, and so that it is unnecessary to do multiple times of tensor importance evaluation and selection within one epoch. The possible variations of tensor importances over different epochs, on the other hand, is also related to the scenario of LLM fine-tuning and the corresponding characteristics of training dataset.
>
> We agree with the reviewer that, if the LLM fine-tuning uses a fixed training dataset, it is possible that using a fixed tensor selection decided at the initial phase of training may not result in a significant model accuracy drop, compared to runtime tensor selection. However, in practical LLM fine-tuning scenarios, this assumption usually does not hold due to the following two reasons. First, a lot of LLM fine-tuning scenarios, such as online learning and model personalization, is continuously training the model using online data, which is continuously being generated at runtime with varying data distributions. Such varying data distributions in training will surely result in different importances of tensors through the training procedure and hence require runtime tensor selection. Such online LLM fine-tuning scenarios is recently getting more and more popular, especially with the possibility of deploying LLMs onto user's personal mobile devices such as smartphones. Second, even for a fixed training dataset, it is also possible that the importances of some tensors may change as the training progresses. In these cases, dynamic tensor selection could improve the trained model accuracy. To verify this, we conducted additional experiments using OPT-2.7B model on the webquestions dataset and generative QA task. The results are as follows:
>
> |                                                        | Model accuracy | Training time |
> |--------------------------------------------------------|----------------|---------------|
> | Fixed tensor selection only in the first epoch of training              | 27.4%          | 0.49h         |
> | Dynamic tensor selection, once in each epoch                 | 28.4%          | 0.5h          |
> | More frequent tensor selection (5 times in each epoch) | 27.5%          | 0.64h         |
>
>
> Note that, such improvement of model accuracy would be dataset and model dependent, but these experiment results above demonstrated the necessity of runtime tensor selection. In addition, our experiment results also showed that such tensor importance evaluation and selection indeed incur very little extra computing overhead.
>
> We hope that these clarifications fully addressed your concerns, and we would be more than happy to discuss further. Thank you so much!

---

> > ### Comment · Reviewer_U1j2 · 2023-11-22
> >
> > Thanks to the authors for the update. I would like to raise my score to 6 given the new response.
> >
> > I would encourage the authors to include the results in the updated version.

---

> > > ### Author Response · Authors · 2023-11-22
> > >
> > > Dear Reviewer,
> > >
> > > Thank you so much for raising your review score! Sure, we will make sure to include these results into the camera-ready version of the paper, if the paper is eventually accepted.

---

### Official Review · Reviewer_CgrN · 2023-10-30

**Soundness:** 3 good
**Presentation:** 2 fair
**Contribution:** 3 good
**Rating:** 5
**Confidence:** 4

**Summary:**

In this paper, the authors propose Green-Trainer for reducing the training cost for adapting pre-trained large language models to downstream applications. The idea is to leverage adaptive backpropagation for adaptively selecting the most appropriate set of LLM
tensors for fine-tuning based on their importance and backpropagation cost in training. This is achieved by measuring the tensor importance based on the cumulative gradient changes of its weight updates in training. Also, a new FLOPs model is built for finding the optimal tensor from an exponential number of possibilities. Compared with existing fine-tuning methods, the proposed Green-Trainer can reduce 64% training FLOPs while achieving similar accuracy.

**Strengths:**

1. The problem of reducing the FLOPs when fine-tuning LLM is important. And the authors propose an effective method for achieving this while maintaining the model accuracy.

2. The authors include a clear motivation for conducting adaptive backpropagation to reduce the computation cost of existing fine-tuning methods for LLMs.

3. The authors conduct extensive experiments to demonstrate the effectiveness of the proposed method compared with the baseline methods.

**Weaknesses:**

1. The  novelty of the proposed method is rather limited. In particular, the method for evaluating the tensor importance has already been proposed for neural network pruning. And the authors did not cite any relevant literature here.

2. The writing of the paper is sometimes confusing. For example, in section 3.1, it is not clear how such profiling is done for different layers in LLM. The authors should give some concrete examples of the FLOPs for these layers to better demonstrate the idea of tensor FLOPs profiling.

3. Since the proposed dynamic programming can only find approximate solutions, there is no discussions on the accuracy of the found solution using the proposed approach.

**Questions:**

1. In the experiments, the authors only fine-tune the model for 5 epochs, then it is possible for some fine-tuning methods (eg. FT-Top2) to have the underfitting issue, how about fine-tuning for more epochs?

2. How the running time is measured? There is a lack of hardware specifications.

---

> ### Author Response · Authors · 2023-11-16
> **Response to reviewer's comments and questions**
>
> Dear reviewer,
>
> Thank you so much for your efforts on reviewing our paper. Please find our responses to your comments and questions below.
>
> > In the experiments, the authors only fine-tune the model for 5 epochs ....
>
> We agree that it is possible for FT-Top2 and other baselines to further have higher accuracy if being fine-tuned for more than 5 epochs. However,  these baseline methods generally involve more computations in each epoch but achieve similar or lower model accuracy compared to GreenTrainer. For example, in Table 2, when fine-tuning a BLOOMZ on DialogSum dataset for 5 epochs, GT-0.5 takes 3.1 hours to reach R1=24.9% while FT-Top2 takes 4.6 hours to reach R1=22.1%. This demonstrates that GreenTrainer has better accuracy-compute efficiency. We also did more experiments by fine-tuning the OPT-2.7B model for 10 epochs on the webquestions dataset. The model accuracy for the generative QA tasks is listed below:
>
> |         | Accuracy after 5 epochs | Accuracy after 10 epochs | validation loss in the first 5 epochs |
> |---------|-------------------------|--------------------------|---------------------------------------|
> | FT-Top2 | 6.3%                    | 7.1%                     | [2.907, 2.805, 2.667, 2.626, 2.616]   |
> | LoRA    | 19.6%                   | 21.9%                    | [1.765, 1.672, 1.646, 1.624, 1.620]   |
>
> In comparison, GT-0.5 has an accuracy of 28.7% after 5 epochs of fine-tuning. Since the model accuracy in baselines increases slowly with more epochs, we believe that such advantage of GreenTrainer will hold with more training epochs.
>
> > How the running time is measured? There is a lack of hardware specifications.
>
> We measured the end-to-end fine-tuning time on a Lambda Lab cloud instance with 24 vCPUs and a single H100 80GB GPU. Note that, in all experiments, the measured fine-tuning time covers the entire fine-tuning procedure, include the overhead of our importance evaluation and dynamic programming.
>
> > The novelty of the proposed method is rather limited. In particular, the method for evaluating the tensor importance has already been proposed for neural network pruning. And the authors did not cite any relevant literature here.
>
> We agree that some existing metrics (e.g., SNIP and Fisher information as suggested by Reviewer B8AP), could be adopted into GreenTrainer for tensor importance evaluation. However, these metrics, as designed for neural network pruning, are used to quantify the importance of model weights at their current values, so the less important weights can be set to zeros during inference. In contrast, our proposed tensor importance evaluation is to quantify the importance of performing weight updates on a tensor to the reduction of training loss. In other words, we measure the importance of “tensor update” instead of the importance of the tensor’s current value itself. Therefore, most existing importance metrics being used in neural network pruning would be ineffective in our training scenario.
>
> In addition, additivity is an essential requirement on the importance metric in our problem formulation. However, existing metrics, such as SNIP ($|\partial L/\partial w \cdot w|$) [1] and Fisher information ($|\partial L/\partial w \cdot w|^2$) [2], take absolute or square values in the metric, and they are hence not additive, because updating parameters can be either constructive or destructive to the loss reduction. In contrast, our proposed importance metric is directly derived from the first-order approximation after Taylor expansion of the training loss. Therefore, it retains a moderate level of additivity in low learning rate ($10^{-5}$) regimes for fine-tuning [3].
>
> [1] https://arxiv.org/abs/1810.02340
> [2] https://arxiv.org/abs/2108.00708
> [3] https://dl.acm.org/doi/10.1145/3581791.3596852
>
> We are also aware of other choices of designing such importance metric while retaining additivity, and we have provided experiment results in Table 4 to demonstrate the performance of GT using different tensor importance metrics, such as the magnitude-based metric ($\Delta_w$) and gradients-only metric ($\partial L/\partial w$).
>
> > The writing of the paper is sometimes confusing ...
>
> We provided technical details about how to obtain tensor FLOPs for different types of layers (e.g., MHA, LayerNorm, and FFN) in section 3.1. In the revised paper draft, we also added an example about practical profiling in our experiments in Appendix A.3 to help understanding. Please check.
>
> > The proposed dynamic programming can only find approximate solutions ....
>
> DP algorithm can find the optimal solution of tensor selection. However, in order to reduce its computing cost, we downscale $T_{full}$ to $T_q$  in Appendix A.2, which introduces such sub-optimality. We studied the impact of such downscaling in Appendix A.2, and results show that we can always find an value of $T_q$ that achieves on-par model accuracy but restrains the computing cost of DP within 1% of the entire fine-tuning cost.

---

> > ### Author Response · Authors · 2023-11-22
> > **Another follow-up**
> >
> > Dear Reviewer,
> >
> > As the deadline of the discussion phase has been approaching (the end of today), we would like to kindly follow up again and check whether our response fully addressed all your concerns. Please feel free to let us know if you have any additional questions or need any further details about our response. Again, thank you so much for your efforts!

---

> ### Author Response · Authors · 2023-11-20
> **Follow up on our previous response**
>
> Dear Reviewer,
>
> We are writing to kindly follow up with our previous response to your review comments, and would like to kindly enquire whether our response fully answered your questions and addressed your concerns to our paper. We will be more than happy to provide further details and responses for any additional questions you may have. Again, thank you so much for your review efforts and your insightful feedback about our paper!

---

### Official Review · Reviewer_P1Bv · 2023-11-02

**Soundness:** 3 good
**Presentation:** 3 good
**Contribution:** 3 good
**Rating:** 6
**Confidence:** 4

**Summary:**

The paper proposes GreenTrainer, an approach for improved balancing of task quality and cost tradeoff of model finetuning.  The main idea is that only tensors that are important for the downstream task should be trained during finetuning so as to simultaneously minimize any quality loss and computation (FLOPs) costs. The paper further presents an algorithm for identifying important tensors for a downstream task at runtime to ensure adaptivity. The evaluation results show up to 64% reduction in FLOPs cost with little or no quality loss compared to fine-tuning the full model, and up to 4% quality improvement compared to existing FLOPs reduction techniques.

**Strengths:**

Model fine-tuning is an important aspect of deep learning, and so the paper is tackling an important and timely problem in trying to reduce the computation costs of finetuning while preserving task quality.

The idea of identifying important tensors at runtime for the target downstream task is interesting and seems more promising than the existing static approaches. It seems that this would be the right solution for the problem, assuming it could be done efficiently.

The quality and reduction improvements presented in the evaluation are quite impressive.

**Weaknesses:**

The main concern is the vagueness on the efficiency or costs of identifying important tensors during finetuning. This makes it difficult for me to judge whether or not this is a practical approach. The tensor selection is simply claimed to be O(N), but I find this to be insufficient to understand the runtime costs. Specifically, I could not glean the following pertinent information from the draft.

1. What is the e2e finetuning time of GreenTrainer compared to baselines.

2. How frequently are important tensors identified? Is it per iteration, per epoch (as suggested by Figure 1), or some other intervals? What is the frequency for the experimental results?

3. What is the `m` binary vector, or at least the number of important tensors, for the experimental results?

**Questions:**

See weakness.

---

> ### Author Response · Authors · 2023-11-16
> **Response to reviewer's comments and questions**
>
> Dear reviewer,
>
> Thank you so much for your efforts on reviewing our paper. Please find our responses to your comments and questions below. We hope these responses fully address your concerns on our paper, and if so we would highly appreciate if you could reconsider your rating of our paper. Please feel free to let us know if you would like further details about any part of the paper, and we would be more than happy to discuss further. Thank you!
>
> > What is the e2e finetuning time of GreenTrainer compared to baselines.
>
> In general, the end-to-end computing time of standard LLM fine-tuning consists of the following components: 1) doing the forward pass to compute training loss; 2) backpropagating the loss gradients to the trainable tensors and 3) compute the updates to the tensors. In GreenTrainer, we add two extra components: 1) evaluation of tensor importance; 2) dynamic programming to find the optimal tensor selection. Our major contribution in this paper is that we are the first to explicitly incorporate the backpropagation cost, i.e., the components 2) and 3) listed above that are the dominant factor of the fine-tuning cost, into the selection of trainable tensors. In this way, we can ensure that the selected set of tensors can guarantee the required reduction of fine-tuning cost with the minimum impact on the trained model accuracy. In contrast, most existing LLM fine-tuning works adopt indirect objectives, such as the parameter count and per-layer forward pass FLOPs, to guide tensor selection. Without a proper backpropagation cost model, the tensor selection of existing works can lead to either insufficient FLOPs reduction or low accuracy of the fine-tuned model.
>
> In all experiments, we measured the end-to-end computing time of LLM fine-tuning, by including all the components described above for standard LLM fine-tuning and the two extra components we added.
>
> Our experiment results showed that our tensor importance evaluation and DP are both very lightweight and introduce little extra computing overhead. For tensor importance evaluation, it by design incurs only negligible computing overhead, because it only takes one extra training iteration to obtain gradients for all model parameters, as described in Section 3.1 of the paper. To further verify such low overhead, we conducted extra experiments and the results show that, even when we repeatedly perform such tensor importance evaluation in each training epoch at runtime to capture the possible variations of tensor importance at runtime, the overhead of importance evaluation is only <0.2% on SciTLDR dataset and <0.01% on DialogSum dataset, with respect to the entire fine-tuning cost. In Appendix A.1, we also showed that, by involving a smaller set of tensors into training, we can further reduce the amount of memory usage by tensor importance evaluation by another 50%.
>
> Similarly, in the Appendix A.2, we also showed that the computing cost of DP is very low, and such computing costs can be further reduced by exploiting different tradeoffs between the model accuracy and cost. For example, by reducing the resolution of solving the DP, we can further reduce the wall-clock time of running DP by another 100x, with negligible lost on the model accuracy.
>
> > How frequently are important tensors identified? Is it per iteration, per epoch (as suggested by Figure 1), or some other intervals? What is the frequency for the experimental results?
>
> In all experiments, we evaluated tensor importance per epoch. As discussed above, our tensor importance evaluation is very lightweight, and even doing so per epoch produces very little extra computing overhead at runtime.
>
> > What is the m binary vector, or at least the number of important tensors, for the experimental results?
>
> The binary vector $\mathbf{m}$ represents the selection of tensors being involved in fine-tuning: the $i$-th component of the vector $\mathbf{m}$ is valued in [0, 1], as 1 indicates that tensor i is selected to be fine-tuned and 0 indicates otherwise. In our approach, we first evaluate the importances of different tensors at runtime, and then solve $\mathbf{m}$ using DP based on our problem formulation in Eq. (5). Note that the solution to $\mathbf{m}$ is not constant throughout the training procedure, because the importances of tensors may change as the training progresses, and we run DP once per epoch to make sure that the tensor selection (represented by $\mathbf{m}$) always reflect the tensors’ current contribution to training.

---

> > ### Comment · Reviewer_P1Bv · 2023-11-17
> > **Iteration to compute tensor importance**
> >
> > Thanks for clarifications regarding e2e time and runtime cost of computing tensor importance. I have the follow up questions:
> > 1. Since tensor importance is computed once per epoch, when does it occur? Is it at the beginning, middle, or towards the end of the epoch? Also, is it the same time for different epochs?
> > 2. Did you evaluate computing tensor importance multiple times during the epoch?

---

> > > ### Author Response · Authors · 2023-11-18
> > > **Further clarifications about tensor importance evaluation**
> > >
> > > Dear Reviewer,
> > >
> > > Thank you so much for your response, and we are glad to see that most of your concerns have been addressed. Please find our responses to your questions as below:
> > >
> > > > Since tensor importance is computed once per epoch, when does it occur? Is it at the beginning, middle, or towards the end of the epoch? Also, is it the same time for different epochs?
> > >
> > > We evaluate the importances of tensors at the beginning of every epoch. Yes, we take the same approach of importance evaluation for different epochs, and so in every epoch, we evaluate the tensor importance at the beginning of the epoch. In this way, we make sure that each epoch's tensor selection is made based on the up-to-date information of tensor importances.
> > >
> > > > Did you evaluate computing tensor importance multiple times during the epoch?
> > >
> > > Following the previous question, we would like to clarify that we only evaluate the tensor importance once in every epoch, more specifically, in the beginning of every epoch. The reasons are two-fold. First, frequently checking the number of iterations in each epoch to decide whether to re-evaluate tensor importance is very computationally expensive and will significantly slow down the training in real implementations. Second, in practical LLM fine-tuning, we usually use a small batch size (e.g., 4 in our implementation and evaluations) but a large number of iterations in each epoch. As a result, we expect only very small changes of tensor importances over different iterations in the same epoch.
> > >
> > > Due to these reasons, we only evaluate tensor importance once per epoch. Our experiment results show that such runtime evaluation incurs very little extra overhead (<0.2% of the entire fine-tuning cost).
> > >
> > >
> > > We hope that our answers could fully address the reviewer's concerns about our paper. Please feel free to let us know if you have any other questions. Thank you so much!

---

> > > > ### Comment · Reviewer_P1Bv · 2023-11-18
> > > >
> > > > Thanks for the response. However, I feel that it would be interesting to explore whether more frequent tensor importance evaluation would bring accuracy benefits. If you have conducted such an evaluation, please share the results. Nevertheless, my concerns are now satisfied by your response to increase my score. Thanks for sharing this great work.

---

> > > > > ### Author Response · Authors · 2023-11-19
> > > > > **Experiment results on different frequencies of tensor importance evaluation within an epoch**
> > > > >
> > > > > Dear Reviewer,
> > > > >
> > > > > First of all, we are glad that our responses fully addressed your concerns and high appreciate that you raised your review scores on our paper!
> > > > >
> > > > > Regarding your question about "whether more frequent tensor importance evaluation would bring accuracy benefits", we conducted some extra experiments by applying more frequent tensor importance evaluation within each epoch, using OPT-2.7B model on the webquestions dataset and generative QA task. The results are as follows:
> > > > >
> > > > > | Frequency of tensor importance evaluation | Accuracy | Wall-clock training time |
> > > > > |-------------------------------------------|----------|--------------------------|
> > > > > | Every 945 iterations (once per epoch)     | 28.4%    | 0.50 h                   |
> > > > > | Every 600 iterations                      | 28.5%    | 0.54h                    |
> > > > > | Every 400 iterations                      | 28.2%    | 0.56h                    |
> > > > > | Every 200 iterations                      | 27.5%    | 0.64h                    |
> > > > >
> > > > > The results show that:
> > > > >
> > > > > 1) more frequent tensor importance evaluation brings only very small improvement on the task accuracy. Considering the randomness in different training trials, we believe that such accuracy improvement is negligible, and the accuracy could even drop down by 1% when the frequency of evaluation is very high (every 200 iterations). We believe that this is due to accumulation of tensor importance evaluation and tensor selection errors, which stem from the first-order approximation in the tensor importance metric and the approximate solution in DP. Another possible reason is that the tensor importances are calculated over the training dataset, and too frequent tensor importance evaluation may make the training process overfit to the training dataset.
> > > > >
> > > > > 2) The training cost steadily increases with the frequency of tensor importance evaluation. When the interval of evaluation reduces from 945 iterations to 200 iterations, the training time increases by 28%.
> > > > >
> > > > > In summary, performing more frequent tensor importance evaluation within each epoch brings little improvement on the task accuracy but noticeably increase the training cost. We hope that you find our experiment results and analysis reasonable and useful. Thanks again!

---

### Official Review · Reviewer_B8AP · 2023-11-05

**Soundness:** 3 good
**Presentation:** 3 good
**Contribution:** 3 good
**Rating:** 5
**Confidence:** 5

**Summary:**

To address the significant computing costs and environmental impact associated with LLM fine-tuning, this paper proposes "GreenTrainer," a new technique designed to minimize FLOPs through adaptive backpropagation. GreenTrainer selectively fine-tunes the most relevant parts of the LLM based on their importance and the computational cost of their training. Experimental results demonstrate that GreenTrainer can reduce training FLOPs by up to 64% compared to traditional full fine-tuning methods, while maintaining accuracy compared to full-finetuning.

**Strengths:**

This paper addresses an important problem mentioned within it: fine-tuning 10K Llama-13b models can be 6.9 times more expensive than pre-training 175Bs. It is crucial to reduce carbon footprint and energy consumption in order to make LLMs sustainable.

Traditional PEFT methods, such as LoRA and adapter, focus on reducing (learnable) parameters but discuss less about fine-tuning FLOPs and throughput. This work explores ways to actually reduce training costs, and the proposed method allows for flexible trade-offs between computation cost and fine-tuning quality.

The authors thoroughly compare the performance on SciTLDR and DialogSum datasets, demonstrating that GreenTrainer achieves up to 4% higher performance with faster training speed compared to existing schemes. Furthermore, the authors provide detailed analysis by varying importance metrics (Table 4), FLOPs reductions (Table 3), and LLM model sizes (Table 5).

**Weaknesses:**

The justification for the effectiveness of adaptive selection is not clear. Language models (LLMs) are typically over-parameterized, and updating a random set of sparse parameters can achieve similar performance. Therefore, it is important to understand why and how adaptive selection properly selects the parameters that are worth fine-tuning. Baselines such as random selection, Fisher, and SNIP should be discussed to provide a comprehensive analysis.

It should be noted that adaptive selection can introduce additional complexity in the training process. Although the paper presents strategies to reduce this cost (dynamic programming), it still adds a layer of complexity to the training process with a complexity of O(N^2 T). There are concerns that this overhead will grow with the size of the model and soon become non-negligible.

The benchmarks in the paper only include summarization tasks and do not provide a holistic evaluation. To claim that a method reduces FLOPs without compromising accuracy, it is necessary to evaluate it on more benchmarks, such as GLUE and E2E NLG Challenge. Even within the summarization tasks, GreenTrainer does not show superior performance compared to LoRA under the **same latency budget**, as shown in Table 2 for FLAN-T5, where GT-0.4 and LoRA are shows very close performance and throughput.

**Questions:**

It is unclear whether the fine-tuning process includes the DP (data parallelism) overhead or if it is counted separately. It would be helpful to clarify the ratio of DP overhead during fine-tuning.

How X% of FLOPs saving can lead to a X% reduction in latency in Tab 2? LLMs fine-tuning  is  both  computation- and **memory-intensive**. Even with the same FLOPs, the latency can vary significantly depending on how the computation is allocated. For example, updating last 2 blocks may have similar performance with updating a partial set of all blocks, but the latency of latter one is much higher as it back-propagates to the first layer. To provide a more comprehensive understanding, additional benchmark details are needed to justify these observations.

To further evaluate the performance of GreenTrainer, more experiments should be conducted to compare it against baselines such as GLUE (which should be relatively inexpensive to execute) and other tasks like hellaswag, webquestions, and piqa. The community is also interested in the performance on Llama. If the 7B model is causing out-of-memory (OOM) issues, reporting the performance with LoAR and GreenTrainer only, or performing on OpenLlama-3B, would be valuable as well.

---

> ### Author Response · Authors · 2023-11-16
> **Response to reviewer's comments and questions**
>
> Dear reviewer,
>
> Thank you so much for your efforts on reviewing our paper. Please find our responses to your comments and questions below.
>
> > It is unclear whether the fine-tuning process includes the DP (data parallelism) overhead ....
>
> We are not sure whether the reviewer refers DP to dynamic programming (Section 3.3) or data parallelism.
>
> For dynamic programming: all the experiments we did in this paper were evaluating the end-to-end fine-tuning costs, including the extra computations and time needed for both tensor importance evaluation and DP. In Appendix A.2, we provided experiment results of the DP overhead in fine-tuning, which can be flexibly adjusted by controlling its resolution. Our experiment results in Table 7 showed that, by selecting the appropriate DP resolution, we can effectively restrain the DP overhead to be <1% of the entire fine-tuning cost.
>
> For data parallelism: we did our experiments on a single H100 80GB GPU and hence did not involve any communication cost between multiple GPUs for data parallelism.
>
> > How X% of FLOPs saving can lead to a X% reduction in latency in Tab 2? ....
>
> We agree that the amount of FLOPs saving can be different from wall-clock time reduction, due to memory access overhead. In our case, the percentage of such memory access overhead is small due to the strong hardware (H100 80GB GPU), but still noticeable. In Table 1, the percentage of wall-clock time saving in GT-0.4/0.5/0.7 is slightly lower than FLOPs saving, indicating the extra cost of memory access due to tensor importance evaluation and DP. Such difference grows as $\rho$ increases, but is within 5%. In contrast, the memory access overhead in the baselines (e.g., FT-Top2, Prefix-T and LoRA) is generally lower. Even if we include memory access overhead, GT can still achieve higher wall-clock time savings.
>
> The reviewer also mentioned an example that updating the last 2 blocks has similar latency with updating a partial set of all blocks. This is because in OPT and BLOOMZ models the output embedding layer is tied with the input embedding layer. As long as the output layer is selected to be trained, the input layer will be backpropagated for updating, leading to a maximum backpropagation path. In contrast, FLAN-T5 doesn’t tie input and output embeddings, and so FT-Top2’s amount of savings on FLAT-T5 is higher.
>
> > More experiments should be conducted to compare it against baselines ....
>
> We conducted experiments using OPT-2.7B, on webquestions and piqa datasets for generative QA tasks. We evaluate the sentence-level accuracy which requires the generated answer to exactly match the ground truth. Our results are as follows:
>
> | piqa  | Accuracy | PFLOPs | Time  |
> |--------|----------|--------|-------|
> | GT-0.5 | 59.2%    | 130.5  | 4.69h |
> | LoRA   | 49.5%    | 174.0  | 6.27h |
>
> | webquestions | accuracy | PFLOPs | Time  |
> |--------------|----------|--------|-------|
> | GT-0.5       | 28.7%    | 12.0   | 0.5h  |
> | GT-0.6       | 29.5%    | 14.0   | 0.61h |
> | LoRA         | 19.6%    | 16.0   | 0.55h |
>
> > LLMs are typically over-parameterized, and updating a random set of sparse parameters can achieve similar performance. Baselines such as random selection, Fisher, and SNIP should be discussed.
>
> We agree that LLMs are typically over-parametrized, but selecting random parameter sets usually leads to much slower convergence speed, and we need more training epochs in fine-tuning. In addition, since the correlation between an arbitrarily selected set of parameters and the LLM’s behavior is unknown, it could be highly random and time consuming to find a best set of parameters that  maximizes accuracy.
>
> We provided experiment results in Table 4 about the performance of GT using different tensor importance metrics, such as the magnitude-based metric ($\Delta_w$) and gradients-only metric ($\partial L/\partial w$). SNIP ($|\partial L/\partial w * w|$) and Fisher information ($|\partial L/\partial w \cdot w|^2$) are used in pruning to quantify the weight importance at current values. In contrast, our tensor importance evaluation is to quantify the importance of performing weight updates to the reduction of training loss ($-\partial L/\partial w \cdot \Delta w$). In other words, we measure the importance of “tensor update” ($\Delta w$) instead of the importance of the tensor’s current value ($w$).
>
> Furthermore, our metric is directly derived from the first-order approximation after Taylor expansion of the training loss. Hence, it retains a moderate level of additivity in low learning rate ($10^-5$) regimes for fine-tuning.
>
> > There are concerns that this overhead of DP will grow with the size of the model and soon become non-negligible ...
>
> In appendix A.2, we provided experiment results about the computing cost of our proposed DP approach, and showed that, by adjusting the resolution of DP, we can restrain the computing cost of DP within 1% of the entire fine-tuning cost, without affecting accuracy.

---

> ### Author Response · Authors · 2023-11-20
> **Follow up on our previous response**
>
> Dear Reviewer,
>
> We are writing to kindly follow up with our previous response to your review comments, and would like to kindly enquire whether our response fully answered your questions and addressed your concerns to our paper. We will be more than happy to provide further details and responses for any additional questions you may have. Again, thank you so much for your review efforts and your insightful feedback about our paper!

---

> ### Comment · Reviewer_B8AP · 2023-11-21
>
> After reading the authors' reponse (to me and other reviewers), some of my questions are addressed. However, I am showing concerns
>
> * ** restrain the DP overhead to be <1% of the entire fine-tuning cost**
>
> As mentioned in the reponse to #P1Bv, the DP is performed per epoch and this overhead makes sense. However, since the DP search is only performed once per epoch and the gradient / weight may vary a lot during fine-tuning. Does this suggest that GreenTrainer cannot handle large dataset (there are a lot of iterations in one epoch).
>
> * **the percentage of such memory access overhead is small due to the strong hardware (H100 80GB GPU)**
>
> Does this claim limit the applibility of "Green" Trainer? Means that the training can be green and cheap only when you have fancy H100 hardware?
>
> * **We conducted experiments using OPT-2.7B, on webquestions and piqa datasets for generative QA tasks**
>
> This PIQA accuracy looks weird. According to original OPT paper, Appendix A, the lowest performance of OPT is near 62.5 but here the LoRA accuracy is only 49.5%, which is a huge drop.

---

> > ### Author Response · Authors · 2023-11-21
> > **Further clarifications**
> >
> > Dear Reviewer,
> >
> > Thank you so much for your response, and we are glad to see that most of your questions and concerns have been addressed. Please find our responses to your questions as below:
> >
> > > As mentioned in the response to #P1Bv, the DP is performed per epoch and this overhead makes sense. However, since the DP search is only performed once per epoch and the gradient / weight may vary a lot during fine-tuning. Does this suggest that GreenTrainer cannot handle large dataset (there are a lot of iterations in one epoch).
> >
> > First, as we mentioned in our response to reviewer P1Bv, in most of practical LLM fine-tuning (as well as our experiments), a small batch size is usually used and correspondingly there will be a large number of iterations in each epoch. In most of such cases, we expect only very small changes of tensor importances over different iterations in the same epoch. Hence, doing DP (i.e., tensor selection) once in each epoch could ensure its accuracy, even over large datasets that result in many iterations in each epoch. Such accuracy was actually verified in our experiment results over the DialogSum dataset, which is a large-scale dialogue summarization dataset with >13k dialogues.
> >
> > Second, we agree with the reviewer that in few cases, the tensor importances, calculated from the model gradient changes, could exhibit non-negligible differences within one epoch. In these cases, the flexible design of GreenTrainer will allows us to adaptively increase the frequency of tensor importance evaluation and the corresponding DP-based tensor selection (by defult, we will reselect the tensors each time when the tensor importances are being re-calculated). To demonstrate the impact of such more frequent tensor importance evaluation and DP-based tensor selection, we conducted some extra experiments using OPT-2.7B model on the webquestions dataset and generative QA task. The results are as follows:
> >
> > | Frequency of tensor importance evaluation | Accuracy | Wall-clock training time |
> > |-------------------------------------------|----------|--------------------------|
> > | Every 945 iterations (once per epoch)     | 28.4%    | 0.50 h                   |
> > | Every 600 iterations                      | 28.5%    | 0.54h                    |
> > | Every 400 iterations                      | 28.2%    | 0.56h                    |
> > | Every 200 iterations                      | 27.5%    | 0.64h                    |
> >
> > From these results, we can see that more frequent tensor importance evaluation and DP brings very small improvement on the task accuracy. The corresponding increase of training overhead is also limited.
> >
> > > Does this claim limit the applicability of "Green" Trainer? Means that the training can be green and cheap only when you have fancy H100 hardware?
> >
> > We would like to clarify that our design of GreenTrainer targets the scenario where a large amount of users use the **centralized cloud computing resources** (e.g., Infrastructure-as-a-service) for LLM fine-tuning. This is expected to be the major method for end users to do LLM fine-tuning and personalization at present and in the future, and there is a great need for Green AI in this scenario where centralized use of a large amount of powerful computing hardware (e.g., thousands of H100 GPUs) creates very high energy consumption and carbon footprint. Comparatively, the carbon footprint caused by LLM fine-tuning at individual users using low-end computing hardware (e.g., workstation-grade GPUs such as RTX 3xxx/4xxx series) would be negligible. Essentially, the memory bandwidth at today's workstation-grade GPUs is also high and comparable to that of high-end server-grade GPUs. For example, the memory bandwidths of a H100 GPU (3TB/s) and RTX 4090 GPU (1.1TB/s) are at the same magnitude. The training time, instead, mainly depends on the amount of computing power.
> >
> > > This PIQA accuracy looks weird. According to original OPT paper, Appendix A, the lowest performance of OPT is near 62.5 but here the LoRA accuracy is only 49.5%, which is a huge drop.
> >
> > The reason for this accuracy gap is that the way we use the OPT model to generate answers is more challenging than the setup in the original OPT paper. The original OPT paper shares the same evaluation setup for QA tasks with the GPT-3 paper (See 3.1 of [a]). According to sec 2.4 in the GPT-3 paper [a], it formulates the piqa task as a multi-choice QA task where the answer is drawn from a small and predefined candidate set (e.g., ["0", "1"]), by comparing the probability scores only over the candidate tokens. In comparison, we strictly cast the problem to open-ended generation, where the candidate set is unknown. In that case, generating correct answers can be more difficult, because the model could generate totally irrelevant answers and increase its chance of making mistakes.
> >
> > [a] https://arxiv.org/pdf/2005.14165.pdf

---

> > ### Author Response · Authors · 2023-11-22
> > **Another follow-up**
> >
> > Dear Reviewer,
> >
> > As the deadline of the discussion phase has been approaching (the end of today), we would like to kindly follow up again and check whether our recent response fully addressed all your new concerns. Please feel free to let us know if you have any additional questions or need any further details about our response. Again, thank you so much for your efforts!

---

### Author Response · Authors · 2023-11-16
**Summary of responses and paper revisions**

We highly appreciate all the reviewers' insightful comments, feedback and suggestions on our paper. We summarize our responses to the reviewers' questions and comments as below:

• We clarified the overhead of GreenTrainer in aspects of data parallelism, dynamic programming, memory access, and tensor importance evaluation. We also discussed the overall GPU memory consumption when adopting our GreenTrainer methods, and referred to the paper about our experiment results on such memory consumption and proposed techniques to further reduce such memory consumption.

• We clarified the difference between our tensor importance metric and the existing metrics being used in neural network pruning, and highlighted the unique contribution of our proposed tensor importance metric in reducing the cost of LLM fine-tuning. We also highlighted our GreenTrainer’s advantage over several existing methods of resource-efficient fine-tuning.

• We provided justifications and explanations to the reviewer’s concerns, including 1) possible underfitting of existing baseline methods, 2) confusions about our problem formulation, 3) details about profiling FLOPs of tensors in fine-tuning, 4) runtime variations of tensor importance, and 5) the possibility of integration with LoRA.

• We added additional experiment results of 1) the performance comparison between GreenTrainer and LoRA on the webquestions and piqa datasets for generative QA tasks, suggested by reviewer B8AP; 2) the computing overhead of tensor importance evaluation in GreenTrainer, suggested by reviewer P1Bv; 3) the performance comparison between GreenTrainer, LoRA and FT-Top2 with more fine-tuning epochs, suggested by reviewer CgrN.

Please check our responses to each individual reviewer for details. We have also revised the paper draft to reflect the key discussions described above. Again, thank you so much for all the efforts, and please feel free to let us know if you have any further questions or would like to know more details about our paper, and we would be more than happy to discuss further.

---

### Meta-Review · Area_Chair_dZCn · 2023-12-21

**Metareview:**

This paper addresses the cost of training large models by reducing the FLOPs required, focusing on backpropagation. In particular, an adaptive backpropagation method fine-tunes the most relevant parts of the LLM, where relevance is based on importance (to the loss) and cost. Experiments are shown to significantly reduce computation while maintaining accuracy.

  Given the rise of large models, all of the reviewers founds the motivation for the paper important and clear, distinct from most parameter-efficient methods that largely focus on the number of parameters/memory. There were a number of concerns raised, however, including better evidence that the selection method is important compared to naive/random selection and other prior importance selection methods, the complexity introduced in the training process, limitation of evaluation in that it did not contain a diversity of tasks, need for better/clearer efficiency analysis, and selection of when the method is applied to compute the importance (e.g. once per epoch). Through an extensive back and forth with the reviewers, two of the reviewers raised their scores while the other two still rated it as a 5 (slightly below acceptance).

  After looking over the paper, reviews, rebuttal, and discussion, I believe that the paper warrants acceptance. The view of focusing on dynamic backpropagation, paired with the notion of selecting importance based on weight updates to reduce the loss, is an interesting approach to a very important problem. As a result, it would be both highly of interest to the community and the paper is well-executed. While the paper still has some slightly below acceptance, the rebuttals seem to address the larger concerns of these reviewers.

  I highly recommend that the authors take into account the reviews and discussion and add several of the additional experiments and arguments into the paper to strengthen it.

**Justification For Why Not Higher Score:**

Overall, this paper has good relevance/importance to the community, so it could be a higher score just based on that. However, given the ratings it does not seem to justify spotlight.

**Justification For Why Not Lower Score:**

This paper could be downgraded to a reject, as it is borderline based on scores. However, looking at the discussion it seems clear that the major concerns were well-addressed by the authors, even if the reviewers did not end up changing their scores.

---

### Decision · Program_Chairs · 2024-01-16

Accept (poster)